# Distilling SNN Students from ANN Teachers via Spiking Neural Architecture Search

## Abstract

Bridging the performance gap between Spiking Neural Networks (SNNs) and Artificial Neural Networks (ANNs) under low timesteps remains a critical challenge in the SNN community. Recent work uses either ANN-supervised training or automated architecture design to narrow the gap. However, the combination of ANN-supervised training and SNN architecture search remains unexplored, leaving room for further improvement of SNN performance. To address this, we propose Distilling SNN Students from ANN Teachers via Spiking Neural Architecture Search (DSAS) method. It is a training-free spiking neural architecture search method that leverages pre-trained ANN teachers to discover efficient and high-performance SNNs with few timesteps. Specifically, DSAS employs an evolutionary neural architecture search guided by two novel metrics, i.e., Multi-layer Activation Similarity (MAS) and Threshold-guided Gradient Similarity (TGS). MAS aligns ANN and SNN feature maps, yet TGS ensures gradient alignment while tuning spiking activation thresholds. Experiments demonstrate that DSAS achieves state-of-the-art accuracy with four timesteps on both convolution-based and transformer-based search space, effectively narrowing the performance gap of ANN and SNN. For example, DSAS discovers architectures that achieve 65.50% top-1 accuracy on Tiny-ImageNet and 81.97% on CIFAR-100. [1]

## 1 Introduction

Spiking Neural Network (SNN), in recent years, has shown its energy-efficient ability across various tasks, including image classification (Zheng et al., 2021), robot control (Li et al., 2022), and object identification (Kim et al., 2020). This low-power characteristic of SNN primarily stems from its computational paradigm, i.e., SNNs transmit information through event-driven binary spike signals (Roy et al., 2019). However, the binary spikes transmit less information than the counterpart Artificial Neural Network (ANN) (Guo et al., 2024a), which uses floating-point in computation. Thus, the performance of SNN models still has gaps compared to the ANN models (Hao et al., 2023; Luo et al., 2024). To address this, a straightforward solution involves additional temporal information to approximate the floating-point of ANNs, i.e., increasing the timesteps $T$ of SNNs. For example, Rueckauer et al. (2017) use 550 timesteps to convert an ANN-based Inception-V3 model to its SNN counterpart, whereas Lee et al. (2020) adopt 100 timesteps to directly train the SNNs using spike-based backpropagation.

However, lengthy timesteps of SNNs results in slow inference speeds and increased power consumption (Yao et al., 2023). Thus, researchers aim to develop high-performance SNNs with few timesteps by designing novel learning algorithms and model architectures. First, learning algorithms incorporate more external guidance to improve the training effectiveness of SNNs models. For example, Xu et al. (2023) improve SNN performance by Knowledge Distillation (KD) from a pre-trained ANN teacher and find that SNN students vary in how well they learn from the ANN teacher. Second, in terms of model architectures, early efforts focused on manually designing high-performance SNNs, such as the spiking residual architecture (Fang et al., 2021a), and the spiking normalization architecture (Zheng et al., 2021). More recently, spiking neural architecture search methods have emerged to automatically discover novel and high-performing SNN models (Na et al., 2022; Kim et al., 2022;

---

[1]Available code: `https://anonymous.4open.science/r/DSAS-5764`

Liu et al., 2024; Song et al., 2025). This leads to a natural question: ***Can the rich knowledge of ANN teachers effectively guide the discovery of high-performance SNN students?***

Hence, we propose a spiking neural architecture search method, termed DSAS, to automatically discover high-performance SNN student models guided by pre-trained ANN teachers. Specifically, to efficiently explore the search space of SNN students, we introduce a training-free spiking neural architecture search framework based on evolutionary algorithms, inspired by Dong et al. (2023). Furthermore, to effectively identify high-performing SNN candidates from the predefined search space, we propose two evaluation metrics that guide the search process using only a single forward and backward computation. Our contributions are summarized as follows:

- We propose a Multi-layer Activation Similarity (MAS) metric to effectively evaluate the feature similarity between the ANN teacher and the SNN student, which can align intermediate feature representations and improve overall performance.
- We introduce a Threshold-guided Gradient Similarity (TGS) metric to capture the similarity of the gradient direction and adaptively calibrate the firing thresholds of SNN neurons. This design can enable SNN to better mimic the backpropagation dynamics of ANN teacher.
- We experimentally show that DSAS achieves state-of-the-art accuracy with only four timesteps on four static and two neuromorphic datasets. For example, on Tiny-ImageNet, we achieve a top-1 accuracy of 65.50% (>2.34% compared to other methods).

## 2 RELATED WORKS

### 2.1 SPIKING KNOWLEDGE DISTILLATION

Spiking Knowledge Distillation (SKD) is a knowledge distillation technique tailored for SNNs, which can leverage the knowledge of the pre-trained ANN teachers to improve the performance of SNN students. In contrast to traditional knowledge distillation, where both teacher and student models are ANNs, SKD faces a main challenge due to the fundamentally different activation patterns between ANNs and SNNs (Guo et al., 2023; Xu et al., 2023). Specifically, the outputs of SNN neurons are the binary spikes, yet ANN neurons produce continuous floating-point outputs (Deng et al., 2020). Thus, SNN students struggle to effectively learn from ANN teachers.

Existing SKD methods focus on addressing this challenge, which can be divided into homogeneous and heterogeneous SKD methods. The former one constrains the ANN teachers and SNN students to have the same architecture. For example, Joint A-SNN (Guo et al., 2023) proposes a multi-branch structure to effectively distill the knowledge from ANN models. SAKD (Qiu et al., 2024) incorporates temporal dynamics into the loss function to improve the distillation efficiency. BKDSNN (Xu et al., 2024b) designs an adaptive layer between ANN teachers and SNN students to enhance distillation performance. Sparse-KD (Xu et al., 2024a) optimizes the connection of SNN students during the distillation process, resulting in higher efficiency. ESKD (Yang et al., 2025) uses the intermediate feature of ANN models to improve the performance of the distilled SNNs.

In contrast, heterogeneous SKD methods offer greater flexibility by allowing different architectures between ANN teachers and SNN students. For instance, KDSNN (Xu et al., 2023) allows SNN students to have shallow layers with small model sizes. Also, KDSNN (Xu et al., 2023) highlights that the different architecture of SNN students can significantly affect the final distillation performance. This means that the architectures of the high-performance SNN student may be different from the ANN teacher. Our preliminary experiments confirm this: although SNN-ResNet34 is theoretically superior, it underperforms SNN-ResNet18 in practice when both are distilled from ANN-ResNet50 (see **Appendix** A). Therefore, this study aims to effectively, efficiently, and automatically design the high-performance architectures of SNN students that align well with ANN teachers.

### 2.2 SPIKING NEURAL ARCHITECTURE SEARCH

Spiking Neural Architecture Search (SNAS) aims to automatically design SNN architectures with minimal or no expertise. Existing SNAS methods can be categorized into three categories based on their search strategies, i.e., random-based, gradient-based, and evolutionary-based strategy. More specifically, SNASNet (Kim et al., 2022) represents a training-free SNAS approach that utilizes a

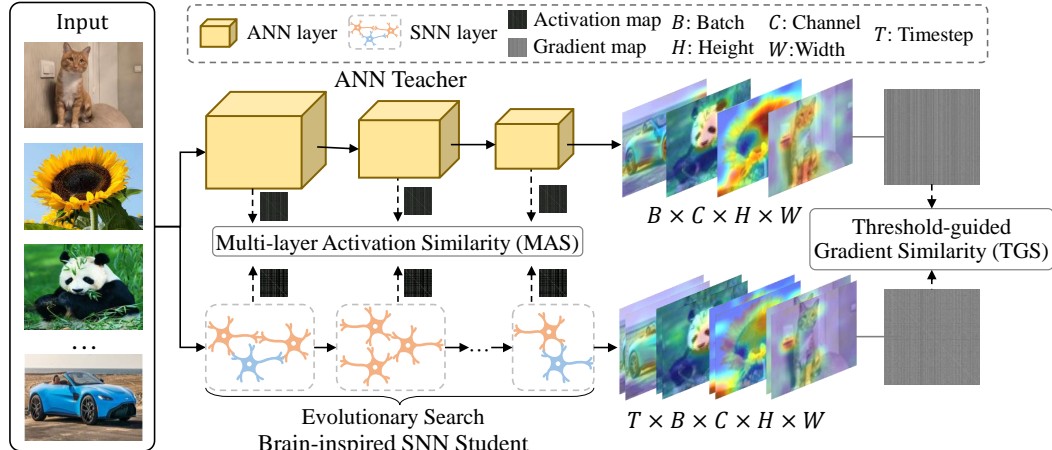

Figure 1: The overall framework of DSAS. The ANN teacher is a pre-trained model, and the high-performance SNN students are automatically searched by the guidance of ANN teacher. The search process only needs one forward propagation for MAS and one backward propagation for TGS.

random search strategy. Among gradient-based methods, notable examples include SpikeDHS (Na et al., 2022), LitE-SNN (Liu et al., 2024), and ESNNs (Yan et al., 2024). SpikeDHS is the first to integrate surrogate gradient search into SNAS. ESNNs aim to enhance the search efficiency of gradient-based SNAS, while LitE-SNN further extends the gradient-based strategy to develop lightweight and efficient SNNs.

As a bio-inspired strategy, evolutionary search matches the brain-inspired nature of SNNs and has been widely used in SNAS methods. For example, AutoSNN (Na et al., 2022) employs an evolutionary search strategy to identify SNN architectures with optimal spike-aware fitness. Additionally, MSE-NAS (Liu et al., 2024) focuses on designing a multi-scale evolutionary search space and a brain-inspired performance predictor. Furthermore, EMO-SNAS (Song et al., 2025) proposes a multi-objective evolutionary search strategy that can design high-performance and energy-efficient SNN architectures. Although the performance of the SNNs designed by existing SNAS methods has surpassed that of manually designed SNNs, a performance gap remains when compared to ANN models. To address this, this study introduces high-performance pre-trained ANN models to guide the search process, which can significantly improve the performance of SNN models.

## 3 METHODOLOGY

### 3.1 OVERALL FRAMEWORK

The overall framework of DSAS is illustrated in Figure 1. High-performance SNN students are searched by the guidance of a pre-trained ANN teacher. The process comprises the following stages: First, a batch of input images is processed by a pre-trained ANN teacher model, which extracts both hierarchical activation patterns and the gradient pattern. Then, the same types of patterns are obtained from each candidate model, i.e., the SNN student, using the same way. After that, to effectively evaluate the similarity between teacher-student pairs, DSAS introduces two metrics, i.e., Multi-layer Activation Similarity (MAS) and Threshold-guided Gradient Similarity (TGS) metrics. More specifically, MAS assesses the similarity between intermediate feature activations, yet TGS measures the similarity between the gradient of the ANN and the SNN. The details of the two metrics are discussed in the next two sections. Finally, the architecture of the SNN student is automatically optimized via evolutionary search, and these two similarity metrics serve as the fitness function to guide the search process. Note that we used the identical search space as Song et al. (2025), and the same evolutionary search strategy following Sun et al. (2020) (more details in **Appendix** B).

## 3.2 MULTI-LAYER ACTIVATION SIMILARITY

As shown in Figure 1, Multi-layer Activation Similarity (MAS) is used to evaluate the activation patterns of the intermediate layers between the teacher-student pairs. Specifically, we calculate the activation similarity of the first, medium, and last layers and the calculation process for each layer is the same as follows. For an ANN teacher and an SNN student with the activation pattens $A_T \in \mathbb{R}^{B \times C \times H \times W}$ and $A_S \in \mathbb{R}^{T \times B \times C \times H \times W}$, where $T$ is the number of timesteps, $B$ is the batch size, and $C$, $H$, and $W$ represent the number of channels, height, and width, respectively.

First, $A_S$ is accumulated at temporal domain to get the fire rate. The process can be summarized as $\bar{A}_S = \sum_{t=1}^{T} A_S^{(t)}$, where $\bar{A}_S \in \mathbb{R}^{B \times C \times H \times W}$. Next, $A_T$ and $\bar{A}_S$ is flatten at the channel, height, and width dimension to generate $A'_T \in \mathbb{R}^{B \times N}$ and $\bar{A}'_S \in \mathbb{R}^{B \times N}$, respectively, where $N = C \cdot H \cdot W$. After that, the sparsity $\rho$ of the student activation is computed using Equation (1):

$$\rho = \frac{1}{BN} \sum_{b=1}^{B} \sum_{n=1}^{N} \mathbf{1} \left[ \bar{A}'_S(b,n) \neq 0 \right], \tag{1}$$

where $\mathbf{1}[\cdot]$ denotes the indicator function. To match this sparsity in the teacher activation, we compute a threshold $\tau$ following Equation (2):

$$\frac{1}{BN} \sum_{b=1}^{B} \sum_{n=1}^{N} \mathbf{1} \left[ A'_T(b,n) > \tau \right] = \rho. \tag{2}$$

We then sparsify the teacher activation patten $A_T$ by setting values less than or equal to $\tau$ to zero, as illustrate in Equation (3):

$$\bar{A}'_T = \begin{cases} A'_T(b,n), & \text{if } A'_T(b,n) > \tau, \\ 0, & \text{otherwise.} \end{cases} \tag{3}$$

We define the similarity matrices for the student and teacher as indicated in Equation (4):

$$\mathcal{M}_T = \frac{(\bar{A}'_T) \cdot (\bar{A}'_T)^\top}{\|(\bar{A}'_T) \cdot (\bar{A}'_T)^\top\|_2}, \quad \mathcal{M}_S = \frac{(\bar{A}'_S) \cdot (\bar{A}'_S)^\top}{\|(\bar{A}'_S) \cdot (\bar{A}'_S)^\top\|_2}, \tag{4}$$

where $\mathcal{M}_T$ and $\mathcal{M}_S$ are the similarity matrices of the ANN teacher and SNN student, respectively, and $\|\cdot\|_2$ denotes the Euclidean norm.

Finally, the similarity function $\mathcal{F}_{\text{MAS}}$ is defined as the mean squared error between the two similarity matrices, as shown in Equation (5):

$$\mathcal{F}_{\text{MAS}} = \frac{1}{B^2} \|\mathcal{M}_T - \mathcal{M}_S\|_2^2. \tag{5}$$

The reason that we propose the multi-layer activation similarity is to improve the accuracy of the high-performance SNN student selection. More specifically, the firing rate of each layer decreases progressively with increasing depth in SNNs, i.e., the deeper layers have sparser activations. In contrast, the activation patterns in the ANN teacher are nearly non-sparse, with sparsity approaching 0%. Thus, if activation similarity is computed only at the last convolutional layer between student and teacher models, shallow SNNs with only one or two layers are more likely to be selected, despite their generally poor performance (see Section 4.5). This is because the similarity between floating-point activations in ANNs and the binary activations of shallow SNNs is high. To address this, we design the sparsity alignment between student-teacher pairs, as described above.

## 3.3 THRESHOLD-GUIDED GRADIENT SIMILARITY

As illustrated in Figure 1, Threshold-guided Gradient Similarity (TGS) is designed to measure the similarity between the gradients of the ANN teacher and the SNN student. This is because gradients determine the convergence direction of the neural network models. Therefore, the more similar the gradients between the teacher and student models, the more effectively the SNN student can learn from the ANN teacher. More specifically, the computation of TGS involves two stages: **(1)** adjusting the firing threshold of SNN neurons under the guidance of the ANN teacher, and **(2)** calculating the gradient similarity between the SNN student and the ANN teacher.

For stage **(1)**, the gradient of each ANN layer is obtained and represented as $\mathcal{G} = \{\mathbf{g}_1, \mathbf{g}_2, \ldots, \mathbf{g}_N\}$, where each $\mathbf{g}_i \in \mathbb{R}^{n_i}$ denotes the gradient corresponding to the $i$-th layer. The gradients $\mathcal{G}$ are then divided into $K$ groups to get the $\alpha$ parameter of Sigmoid$(\cdot)$, where $K$ is a hyperparameter controlling the number of gradient partitions, as illustrated in Equation (6):

$$\mathcal{G} = \bigcup_{k=1}^{K} \mathcal{G}_k, \quad \mathcal{G}_k = \left\{ \mathbf{g}_i^{(k)} \right\}. \tag{6}$$

For each $\mathcal{G}_k$, all elements within the group are treated as a single high-dimensional vector, denoted as $\mathbf{v}^{(k)} = \left( g_1^{(k)}, g_2^{(k)}, \ldots, g_m^{(k)} \right) \in \mathbb{R}^{d_k}$, where $d_k$ represents the total dimensionality of the group $k$. Each $\mathbf{v}^{(k)}$ is then normalized, as shown in Equation (7):

$$\tilde{\mathbf{v}}^{(k)} = \frac{\mathbf{v}^{(k)} - \min(\mathbf{v}^{(k)})}{\max(\mathbf{v}^{(k)}) - \min(\mathbf{v}^{(k)}) + \varepsilon}, \tag{7}$$

where $\varepsilon > 0$ is a small constant introduced to prevent division by zero.

After that, the mean $\mu_k$ and standard deviation $\sigma_k$ of $\tilde{\mathbf{v}}^{(k)}$ are computed. Then, a signal-to-noise-like metric for each group can be defined following Equation (8):

$$\tilde{\alpha}_k = \left( 2 - \frac{\mu_k / (\sigma_k + \varepsilon)}{\max_j \left( \mu_j / (\sigma_j + \varepsilon) \right)} \right) \cdot \alpha_0, \tag{8}$$

where the maximum $\max_j(\cdot)$ is taken over all $j = 1, 2, \ldots, K$ and $\alpha_0$ is a predefined scaling factor. As a result, we obtain a set of scalar coefficients $\{\tilde{\alpha}_1, \tilde{\alpha}_2, \ldots, \tilde{\alpha}_K\}$. These coefficients can encode the statistical properties of the gradients of the ANN teacher across different parts of the network. Finally, these coefficients are used as the $\alpha$ parameter of the surrogate function Sigmoid$(\cdot)$.

The design motivation of the stage **(1)** is from the Signal-to-Noise Ratio (SNR) (Czanner et al., 2015) of the information and behavior of the Sigmoid function. First, SNR serves as a proxy to measure the stability of gradient across different layers of the ANN teacher network. The higher the SNR value, i.e., the smaller $\tilde{\alpha}_k$, indicates the more stable gradient information distribution. Second, the sharpness of the surrogate gradient of the Sigmoid function is significantly influenced by the $\alpha$ parameter. A smaller $\alpha$ value results in a smoother and more stable surrogate gradient (more details in **Appendix** D). Thus, we assign the computed $\tilde{\alpha}_k$ values from the ANN teacher as the $\alpha$ parameter of the SNN student, thereby aligning the gradient distributions between the teacher-student pair.

In stage **(2)**, the gradient similarity between the ANN teacher and SNN student is computed. Specifically, we use the gradients of the final layer from the teacher-student pair, denoted as $G_T \in \mathbb{R}^{N \times C_1}$ and $G_S \in \mathbb{R}^{N \times C_2}$, respectively. Here, $N$ represents the number of classes, while $C_1$ and $C_2$ denote the number of feature channels for the ANN teacher and the SNN student, respectively. Then, the similarity between these gradients is calculated following the method proposed in (Dong et al., 2023), as defined in Equations (9) and (10):

$$\mathcal{P}_T = \frac{(G_T) \cdot (G_T)^\top}{\|(G_T) \cdot (G_T)^\top\|_2}, \quad \mathcal{P}_S = \frac{(G_S) \cdot (G_S)^\top}{\|(G_S) \cdot (G_S)^\top\|_2}, \tag{9}$$

$$\mathcal{F}_{\text{TGS}} = \frac{1}{N \cdot C'} \|\mathcal{P}_T - \mathcal{P}_S\|_2^2, \tag{10}$$

where $C'$ is the minimize value of $C_1$ and $C_2$.

## 4 EXPERIMENTS

### 4.1 EXPERIMENTAL SETTINGS

The experiments are conducted on four static datasets, i.e., ImageNet, Tiny-ImagetNet (Le & Yang, 2015), CIFAR-10, and CIFAR-100 (Krizhevsky et al., 2009), and two neuromorphic datasets, i.e., CIFAR10-DVS (Li et al., 2017) and DVS128-Gesture (Amir et al., 2017), following the conventions of the SNN community (Bu et al., 2022; Kim et al., 2022; Qiu et al., 2024; Song et al., 2024) (details in **Appendix** E.1). Furthermore, the experimental settings of the proposed DSAS method include search settings and training settings. Note that the experimental results of the peer competitors in the table below are cited from the original paper. Note that we utilize 'C-10', 'C-100', 'C10-DVS', and 'Gesture' to denote the acronyms of the CIFAR-10, CIFAR-100, CIFAR10-DVS, and DVS128-Gesture datasets. The RestNet and PyramidNet series are denoted by 'R-' and 'P-'.

**Search Settings.** In the search stage, we adopt the same search space as Song et al. (2025). Specifically, the search space consists of a single type of convolutional layer, varying in the number of channels and stride. In terms of the search strategy settings, we follow the method of Sun et al. (2020), yet with a population size that is $10\times$ larger. More details about the settings for different dataset are provided in **Appendix** E.2.

**Training Settings.** The training process aims to evaluate the performance of the searched architecture. Specifically, the loss function of the knowledge distillation is the commonly used Logits function (Ba & Caruana, 2014), and the weights of the searched SNN students are randomly initialized with Kaiming normalization. More details of the parameter settings are presented in **Appendix** E.3.

## 4.2 OVERALL RESULTS

### 4.2.1 STATIC IMAGE CLASSIFICATION

Table 1: Comparison with the state-of-the-art SNN methods on Tiny-ImageNet. '$\rightarrow$' denotes the spiking knowledge distillation method with 'ANN teacher$\rightarrow$ SNN student'.

| Methods | Architecture | Params (M) | Timesteps | Acc. (%) |
|---|---|---|---|---|
| SSF (Wang et al., 2023) | ResNet-34 | 21.33 | 20 | 58.81 |
| SCA (Li et al., 2024) | VGG-16 | 138.34 | 4 | 49.33 |
| AC2AS (Tang et al., 2023) | VGG-13 | 133.0 | 3 | 54.91±0.20 |
| CLIF (Huang et al., 2024) | VGG-13 | 133.0 | 4 | 63.16 |
| RCS (Wu et al., 2025) | ResNet-18 | 11.22 | 3 | 56.69 |
| Joint A-SNN (Guo et al., 2023) | VGG-16→VGG-16 | 138.34 | 4 | 55.39 |
|  | VGG-16→VGG-16 | 138.34 | 2 | 53.91 |
| Sparse-KD (Xu et al., 2024a) | ResNet-18→ResNet-18 | 11.22 | 4 | 56.92 |
| SNASNet (Kim et al., 2022) | Searched | 74.62 | 5 | 54.60±0.48 |
| AutoSNN (Na et al., 2022) | Searched | - | 8 | 46.79 |
| ESNNs (Yan et al., 2024) | Searched | 6.91 | 3 | 58.59 |
| SpikeNAS-Bench (Sun et al., 2025) | Searched | 75.2 | 5 | 53.6 |
| EMO-SNAS (Song et al., 2025) | Searched | 7.82 | 2 | 54.36 |
| **DSAS (Ours)** | PyramidNet200→Searched | 27.19 | 4 | **65.50** |

Table 2: Comparison with the state-of-the-art SNN methods on ImageNet.

| Methods | Architecture | Params (M) | Timesteps | Acc. (%) |
|---|---|---|---|---|
| Hybrid training (Rathi et al., 2020) | ResNet-34 | 21.79 | 250 | 64.68 |
| QCFS (Bu et al., 2022) | ResNet-34 | 21.79 | 32 | 69.37 |
| Diet-SNN (Rathi & Roy, 2021) | VGG-16 | 138.34 | 5 | 69.00 |
| TET (Deng et al., 2022) | ResNet-34 | 21.79 | 4 | 68.00 |
| STBP-tdBN (Zheng et al., 2021) | ResNet-152 | 60.19 | 4 | 69.20 |
| KDSNN (Xu et al., 2023) | ResNet-50→ResNet-50 | 25.56 | 4 | 67.72 |
| SAKD (Qiu et al., 2024) | ResNet-34→ResNet-34 | 21.79 | 4 | 70.04 |
| BKDSNN (Xu et al., 2024b) | ResNet-50→ResNet-50 | 25.56 | 4 | **72.32** |
| ESKD (Yang et al., 2025) | ResNet-34→ResNet-34 | 21.79 | 4 | 70.64 |
| SpikeDHS (Che et al., 2022) | Searched | 58 | 6 | 68.64 |
| **DSAS (Ours)** | PyramidNet101→Searched | 15.60 | 4 | 68.29 |

**Tiny-ImageNet & ImageNet.** Most existing SNAS methods are verified on Tiny-ImageNet, thus, we first provided the experimental results on this dataset. As shown in Table 1, DSAS significantly outperforms all prior SNN methods, achieving 66.00% accuracy with four time steps and 27.19M parameters. Compared to the spiking knowledge distillation methods, DSAS improves upon the best distillation baseline, i.e., Spare-KD, by 8.58%, while maintaining a reasonable model parameter and a timestep. In the category of searched SNN architectures, all of them struggle to surpass 59% accuracy, even with large model sizes. In comparison, the accuracy of DSAS exceeds the best search-based method, i.e., ESNNs, by nearly 7%. Moreover, we also evaluate DSAS on ImageNet.

Table 3: Compare with SOTA SNN methods on 'CIFAR-10' and 'CIFAR-100'.

| Methods | Architecture | Params (M) | Time -steps | CIFAR-10 Acc.(%) | CIFAR-100 Acc.(%) |
|---|---|---|---|---|---|
| Hybrid training (Rathi et al., 2020) | VGG-11 | 9.27 | 125 | 92.22 | 67.87 |
| QCFS (Bu et al., 2022) | ResNet-20 | 10.91 | 32 | 93.30 | 68.48 |
| Diet-SNN (Rathi & Roy, 2021) | ResNet-20 | 10.91 | 10/5 | 92.54 | 64.07 |
| TET (Deng et al., 2022) | ResNet-19 | 12.63 | 4 | 94.44 | 74.47 |
| STBP-tdBN (Zheng et al., 2021) | ResNet-19 | 12.63 | 4 | 92.92 | 70.86 |
| Joint A-SNN (Guo et al., 2023) | ResNet-18→ResNet-18 | 11.22 | 4 | 95.45 | 77.39 |
| | ResNet-34→ResNet-34 | 21.33 | 4 | 96.07 | 79.76 |
| EnOF-SNN (Guo et al., 2024b) | ResNet-19→ResNet-19 | 12.63 | 2 | 96.19±0.10 | 82.43±0.09 |
| | ResNet-20→ResNet-20 | 10.91 | 2 | 93.86±0.07 | 71.55±0.12 |
| SAKD (Qiu et al., 2024) | ResNet-19→ResNet-19 | 12.63 | 4 | 96.06 | 80.10 |
| BKDSNN (Xu et al., 2024b) | ResNet-19→ResNet-19 | 12.63 | 4 | 94.64 | 74.95 |
| Sparse-KD (Xu et al., 2024a) | ResNet-18→ResNet-18 | 11.22 | 4 | - | 73.01 |
| ESKD (Yang et al., 2025) | ResNet-18→ResNet-18 | 11.22 | 6 | 96.14±0.03 | 79.40±0.16 |
| KDSNN (Xu et al., 2023) | PyramidNet18→ResNet-18 | 11.22 | 4 | 93.41 | - |
| SNASNet (Kim et al., 2022) | Searched | 19.52/20.62 | 5 | 93.73±0.32 | 73.04±0.36 |
| AutoSNN (Na et al., 2022) | Searched | 20.92 | 8 | 93.15 | 69.16 |
| MONAS (Saghand & Lai-Yuen, 2025) | Searched | - | 3 | 93.76 | 73.63 |
| SpikeDHS (Che et al., 2022) | Searched | 14 | 6 | 95.50±0.03 | 76.25±0.10 |
| LitE-SNN (Liu et al., 2024) | Searched | **3.60/3.62** | 6 | 95.60±0.24 | 77.10±0.04 |
| ESNNs (Yan et al., 2024) | Searched | 23.47/27.55 | 3 | 94.64 | 74.78 |
| MSE-NAS (Pan et al., 2024) | Searched | 11.79 | 4 | 95.56 | 77.18 |
| SpikeNAS-Bench (Sun et al., 2025) | Searched | 19.16/20.31 | 5 | 89.3 | 65.2 |
| EMO-SNAS (Song et al., 2025) | Searched | 5.37/3.57 | **2** | 93.60 | 69.59 |
| **DSAS (Ours)** | PyramidNet200→Searched | 12.83/15.78 | 4 | **96.55** **96.50±0.04** | **81.97** **81.73±0.17** |

As indicated in Table 2, DSAS achieves competitive accuracy with only 15.6 M parameters and four timesteps, highlighting its high efficiency. The slightly lower accuracy arises from hardware limits during search stage. More specifically, unlike existing SNAS methods that transfer the architectures searched on CIFAR to ImageNet, DSAS searches directly on ImageNet in just 0.031 GPU-day. However, the GPU memory of the RTX 3090 constrains the model scale for high-resolution images like ImageNet. Additionally, visualizations of the searched architecture and the evolutionary trajectory are provided in **Appendix** F and **Appendix** G, respectively.

Table 4: The performance of SNN student guide by different ANN teachers.

| Dataset | ANN teacher | | | |
|---|---|---|---|---|
| | R-18 | R-34 | R-50 | P-110 |
| C-100 | 74.42 | 79.70 | 79.71 | 80.82 |
| C-10 | 94.73 | 95.22 | 95.18 | 95.90 |

Table 5: Performance of ANN and SNN on various datasets. $^\dagger$/$^\ddagger$ indicate results before/after distillation.

| | C-10 | C-100 | C10-DVS | Gesture |
|---|---|---|---|---|
| ANN Teacher | 97.06 | 82.86 | 82.10 | 97.92 |
| SNN Student$^\dagger$ | 95.11 | 74.90 | 79.20 | 95.49 |
| SNN Student$^\ddagger$ | 96.50 | 81.97 | 81.00 | 96.53 |

**CIFAR-10 & CIFAR-100.** As indicated in Table 3, DSAS performs better on CIFAR-10 and CIFAR-100, reaching accuracies of 96.43% and 81.36%, respectively. Specifically, on CIFAR-10, the method without knowledge distillation has a performance gap compared to DSAS and suffers from high latency. The spiking knowledge distillation methods significantly improve accuracy by leveraging powerful ANN teacher models. However, fixed SNN student models hinder accuracy. For CIFAR-100, the performance improvement of DSAS is significant, e.g., $> 10\%$ better than AutoSNN and EMO-SNAS. Additionally, as shown in Table 4, we investigate the performance of the SNN student guided by different ANN teachers

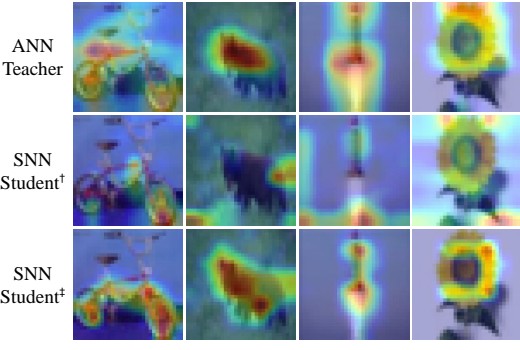

Figure 2: The activation map of ANN and SNN.

Table 6: Compare with SOTA SNN methods on 'CIFAR10-DVS' and 'DVS128-Gesture'.

| Dataset | Methods | Architecture | Params. (M) | Timestep | Acc. (%) |
|---|---|---|---|---|---|
| CIFAR10-DVS | Rollout (Kugele et al., 2020) | DenseNet | **0.5** | 48 | 66.75 |
| | STBP-tdBN (Zheng et al., 2021) | ResNet-19 | 12.63 | 40 | 67.80 |
| | SEW (Fang et al., 2021a) | 7B-Net | 1.19 | 16 | 74.40 |
| | SAKD (Qiu et al., 2024) | ResNet-19→ResNet-19 | 12.63 | 4 | 80.30 |
| | BKDSNN (Xu et al., 2024b) | Wide-7B-Net→Wide-7B-Net | 1.19 | 8 | 72.20 |
| | AutoSNN (Na et al., 2022) | Searched | - | 20 | 72.50 |
| | ESNNs (Yan et al., 2024) | Searched | 3.53 | 10 | 78.40 |
| | **DSAS (Ours)** | PyramidNet110→Searched | 13.83 | **4** | **81.20** |
| DVS128-Gesture | Rollout (Kugele et al., 2020) | DenseNet | 0.8 | 240 | 95.68 |
| | STBP-tdBN (Zheng et al., 2021) | ResNet-19 | 12.63 | 40 | 96.87 |
| | SEW (Fang et al., 2021a) | 7B-Net | **0.13** | 16 | **97.92** |
| | AutoSNN (Na et al., 2022) | Searched | - | 20 | 96.53 |
| | EMO-SNAS (Song et al., 2025) | Searched | 0.47 | 10 | 96.88 |
| | **DSAS (Ours)** | PyramidNet110→Searched | 0.79 | **4** | 96.53 |

and find that DSAS can still achieve great performance. Furthermore, the experimental results indicate that the searched SNN student can gain more accuracy improvements on the complex CIFAR-100 (see Table 5). We also visualize the activation of the ANN teacher and the SNN student before or after distillation in Figure 2. The results indicate that the high-performance distilled SNN student (denoted by '‡') matches the activation map of ANN, yet the SNN student before distillation (denoted by '†') fails to focus on the object. Moreover, as presented in **Appendix** G, the evolutionary progress is more stable than Tiny-ImageNet. Finally, the searched architecture is illustrated in **Appendix** F, which is simpler than Tiny-ImageNet.

### 4.2.2 NEUROMORPHIC IMAGE CLASSIFICATION

**CIFAR10-DVS.** Table 6 presents the comparison of existing SNN methods on DVS-CIFAR10. Among these approaches, DSAS consistently outperforms others. Specifically, DSAS achieves a accuracy of 81.00% with four timesteps. Recent methods based on neural architecture search, such as ESNNs, utilize fewer parameters and run at 10 time steps, yet its accuracy still lags behind DSAS.

**DVS128-Gesture.** As shown in Table 6, DSAS achieves the comparable accuracy of 95.49% with only 0.79M parameters and four timesteps, demonstrating superior efficiency compared to most existing approaches. Although methods such as STBP-tdBN and SEW report slightly higher accuracies, they require significantly more parameters or longer timesteps. Note that the searched architecture and evolution trajectory of neuromorphic datasets are provided in **Appendix** F and G.

### 4.3 PARAMETER STUDY ON TIMESTEPS

This section analyzes DSAS performance across timesteps. As shown in Table 7, accuracy increases with larger timesteps but the improvement slows gradually. Figure 3 further indicates that SNN students with fewer timesteps absorb more knowledge from ANN teachers, achieving greater improvement. This can provide the insight for designing one-step, high-performance SNNs.

Table 7: The image classification accuracy under different timestep.

| Dataset | Timestep | | | |
|---|---|---|---|---|
| | $T = 1$ | $T = 2$ | $T = 4$ | $T = 6$ |
| C-100 | 78.33 | 80.48 | 81.97 | 82.06 |
| C-10 | 94.76 | 95.93 | 96.55 | 96.56 |
| C10-DVS | 76.20 | 79.40 | 81.00 | 81.20 |
| Gesture | 81.59 | 94.10 | 95.83 | 96.88 |

Figure 3: The accuracy with and without KD under different timestep.

## 4.4 Efficacy Study

**Search cost analysis.** As shown in Table 8, DSAS achieves consistently ultra-low search costs across all datasets, including the static large-scale ImageNet and Tiny-ImageNet and the complex neuromorphic datasets. This can demonstrate the superior efficiency of DSAS compared to others. Note that although the search cost for ImageNet increases when the ANN teacher's training time is included, the pretrained weights are easily accessible and reusable on GitHub or Hugging Face.

**Energy efficiency analysis.** As illustrated in Table 9, SNN Student drastically reduces energy compared to ANN Teacher. Distillation slightly increases energy use but significantly boosts SNN performance, demonstrating improved performance with minimal energy impact. The methodology for energy evaluation is provided in **Appendix** H.

Table 8: The search cost (GPU days) of different SNAS methods. '–' means not searched on this dataset. '*' indicates that ANN teacher training time is considered.

| Method | ImageNet | Tiny-ImageNet | CIFAR | C10-DVS | Gesture |
|---|---|---|---|---|---|
| AutoSNN | - | 5.2 | 1.9 | 1.3 | - |
| SpikeDHS | - | - | 1.4 | - | - |
| LitE-SNN | - | - | 5.1 | - | - |
| ESNNs | - | 0.207 | 0.07 | 0.15 | - |
| SNASNet | - | 0.17 | 0.127 | - | - |
| EMO-SNAS | - | 57 | 25.5 | - | 2 |
| **DSAS (Ours)** | 0.031 | 0.044 | 0.088 | 0.052 | 0.005 |
| **DSAS* (Ours)** | 7.351 | 0.478 | 0.471 | 0.422 | 0.189 |

Table 9: Energy cost comparison for a single forward on CIFAR-100 and CIFAR10-DVS. $^{\dagger}/^{\ddagger}$ indicate results before/after distillation.

| Model | Energy (mJ) | |
|---|---|---|
| | C-100 | C10-DVS |
| ANN Teacher | 21.07 | 349.00 |
| SNN Student$^{\dagger}$ | 0.52 | 0.61 |
| SNN Student$^{\ddagger}$ | 0.95 | 1.04 |

**Transferability analysis.** As shown in Table 10, each architecture is searched on the original dataset and directly evaluated on the target dataset with only modification on the classification head. The results show that architectures searched on static or neuromorphic datasets transfer effectively to other datasets, achieving great accuracy and parameters.

**Different search & optimization method analysis.** As indicated in Table 11, the results of the random search are lower than the DSAS. Then, we compare the optimization methods without the ANN teacher. Specifically, multi-objective Evolutionary Computation (EC) (Song et al., 2025) achieves a small model size but lags behind in accuracy compared to our DSAS. Moreover, zero-cost SAHD (Kim et al., 2022) has bigger parameters of 27.18 M but with an inferior accuracy of 78.86% compared to DSAS. Finally, compared to the Bayesian (Akiba et al., 2019) hyperparameter tuning method, DSAS achieves better performance. More discussions are presented in **Appendix** I.

Table 10: Cross-dataset transfer performance of architectures searched by DSAS.

| Original Dataset | Target Dataset | Accuracy (%) | Parameter (M) |
|---|---|---|---|
| Tiny-ImageNet | CIFAR-10 | 96.38 | 27.17 |
| ImageNet | CIFAR-10 | 94.02 | 15.50 |
| CIFAR-10 | Tiny-ImageNet | 65.41 | 38.72 |
| CIFAR10-DVS | CIFAR-10 | 92.97 | 13.84 |
| CIFAR10-DVS | DVS128-Gesture | 95.83 | 13.83 |

Table 11: Performance comparison of different search & optimization methods on CIFAR-100.

| Optimization Method | Nature Acc. (%) | KD Acc. (%) | Parameter (M) |
|---|---|---|---|
| Random Search (best) | 73.50 | 80.87 | 10.68 |
| Random Search (worst) | 71.40 | 77.91 | 26.72 |
| Multi-objective EC | 69.59 | 74.50 | 3.57 |
| Zero-Cost (SAHD) | 74.70 | 78.86 | 27.18 |
| Bayesian Optimization | 53.25 | 60.23 | 16.83 |
| **DSAS (Ours)** | 74.90 | 81.97 | 15.78 |

**Search space of spiking Transformer.** To further justify the effectiveness of DSAS, we utilize the encoder block of Spikformer (Zhou et al., 2023) as the basic search block and search the block number and dimension following Auto-Spikformer (Che et al., 2024) and AutoST (Wang et al., 2024). As shown in Table 12, DSAS achieves the accuracy of 96.50% and 83.86% on CIFAR-10 and CIFAR-100, respectively, which are better than the other methods.

**Search for multiple types of spiking neurons.** To explore the potential of DSAS in optimizing neuron types, we add four types of SNN neurons in the search space, i.e., IF (Gerstner et al., 2014), LIF (Gerstner et al., 2014), PLIF (Fang et al., 2021b), and ELIF (Brette & Gerstner, 2005), and jointly optimize architecture and neuron types. As presented in Table 13, our method leads to competitive results, achieving 96.12% on CIFAR-10 and 81.08% on CIFAR-100.

Table 12: Performance comparison on the Transformer-based search space for CIFAR datasets.

| Method | Architecture | Param. (M) | CIFAR-10 Acc. (%) | CIFAR-100 Acc. (%) |
|---|---|---|---|---|
| Spikformer | Spikformer-4-384 | 9.32 | 95.19 | 77.86 |
| Auto-Spikformer | Searched | 4.69 | 95.23 | 77.91 |
| AutoST | Searched | 11.52 | 96.03 | 79.44 |
| **DSAS (Ours)** | Searched | 12.89 | 96.50 | 83.86 |

Table 13: DSAS performance when jointly optimizing architecture and neuron types on CIFAR.

| Dataset | Acc. (%) |
|---|---|
| CIFAR-10 | 96.12 |
| CIFAR-100 | 81.08 |

## 4.5 ABLATION STUDY

**MAS.** To evaluate the effectiveness of the proposed MAS metric, we analyze both the performance and depth distribution of the searched architectures. As shown in Table 14, the architecture obtained using the MAS metric outperforms that derived from the AS metric on the CIFAR dataset. The AS metric is based solely on the activation of the last convolutional layer, which tends to favor shallow architectures (see Figure 4). In contrast, the MAS metric can effectively address this issue, as shown in Figure 5, the last generation maintains a relatively uniform depth distribution.

**TGS.** The effectiveness study of TGS contains two parts. The first assesses the performance of the searched architecture. As shown in Table 14, TGS achieves higher accuracy than AS. The second part examines the effect of the $\alpha$ value on the performance of the searched architecture. As indicated in Table 15, MAS can improve the performance using the ANN-based $\alpha$.

Table 14: The ablation studies of the proposed MAS and TGS metrics.

| Metric | CIFAR-100 Acc. (%) | CIFAR10 Acc. (%) |
|---|---|---|
| AS | 76.62 | 94.26 |
| **MAS (Ours)** | 77.55 | 95.11 |
| **TGS (Ours)** | 80.75 | 96.13 |
| **TGS+MAS** | 81.63 | 96.55 |

Table 15: The image classification accuracy under different $\alpha$ value of TGS metric.

| Dataset | $\alpha = 2$ | $\alpha = 3$ | ANN-guided $\alpha$ |
|---|---|---|---|
| C-100 | 80.01 | 81.40 | 81.97 |
| C-10 | 95.56 | 96.19 | 96.55 |
| C10-DVS | 80.20 | 80.30 | 81.00 |
| DVS128-G | 88.89 | 96.18 | 96.53 |

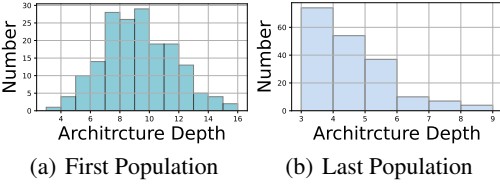

(a) First Population   (b) Last Population

Figure 4: The distribution of the architecture depths under the proposed AS metric.

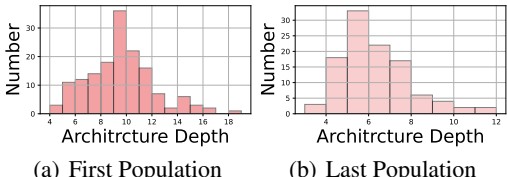

(a) First Population   (b) Last Population

Figure 5: The distribution of the architecture depths under the proposed MAS metric.

## 5 CONCLUSION

In this paper, we propose a training-free spiking neural architecture method guided by the pre-trained ANN teacher. This method is termed DSAS, which can automatically discover the high-performance SNN models with low timesteps. To achieve this, two novel metrics are designed to select the great SNN students during the search process, i.e., Multi-layer Activation Similarity (MAS) and Threshold-guided Gradient Similarity (TGS) metrics. The former one can effectively evaluate the activation patterns between the ANN teacher and the SNN student. The latter one can measure the similarity between teacher-student pairs. Both static and neuromorphic datasets verify the efficacy of DSAS. For example, DSAS can achieve 65.50% on Tiny-ImageNet and 81.97% on CIFAR-100.

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

## A  INFLUENCE OF SNN STUDENT WITH DIFFERENT ARCHITECTURE

This section aims to analyze the impact of different SNN student architectures on the distillation performance. As shown in Table 16, despite its larger capacity, the SNN-ResNet34 distilled from ANN-ResNet50 consistently lags behind SNN-ResNet18 across both CIFAR-10 and CIFAR-100. For example, at $T = 4$, SNN-ResNet34 reaches 93.80% on CIFAR-10 and 73.47% on CIFAR-100, whereas SNN-ResNet18 achieves 94.09% and 77.39%, respectively. This empirical evidence supports our observation that SNN-ResNet18 is more effective in practice, even though SNN-ResNet34 is theoretically more powerful. Moreover, when the ANN teacher changed, the performance of the SNN student was also significantly influenced. For instance, ANN-ResNet34 yields a student (SNN-ResNet18) with 94.55% on CIFAR-10 and 77.13% on CIFAR-100 at $T = 4$, which is on par with or

even slightly better than the counterpart distilled from ANN-ResNet50. This suggests that a larger ANN teacher does not necessarily guarantee better distillation results for SNN students.

In summary, exploring SNN architectures different from the source ANN is useful because a theoretically larger student, e.g., SNN-ResNet34, may underperform a smaller one, e.g., SNN-ResNet18, indicating that optimal SNN architectures do not necessarily align with their ANN counterparts.

Table 16: The accuracy of SNN students with different architectures.

| ANN teacher | SNN Student | CIFAR-10 Acc. (%) | | | CIFAR-100 Acc. (%) | | |
|---|---|---|---|---|---|---|---|
| | | $T = 1$ | $T = 2$ | $T = 4$ | $T = 1$ | $T = 2$ | $T = 4$ |
| ResNet-50 | ResNet-34 | 87.55 | 92.71 | 93.80 | 29.71 | 65.78 | 73.47 |
| | ResNet-18 | 90.45 | 94.30 | 94.09 | 72.69 | 75.52 | 77.39 |
| ResNet-34 | ResNet-18 | 93.29 | 94.30 | 94.55 | 73.17 | 75.19 | 77.13 |

## B  DETAILS OF SEARCH SPACE & SEARCH STRATEGY

This section presents the search space and search strategy used in this study. As for the search space, we utilize the same search space as Song et al. (2025). The search space contains only one type of basic block in the search space, as shown in Figure 6. All the searched architectures are built based on the basic block with different input and output channels. Moreover, to achieve the downsample operation, the convolution stride of the basic block is set to 's=2', as presented in Figure 7. The downsample block is a variant of the basic block, where it only sets the convolution stride of the basic block to two and involves $1 \times 1$ convolution at the skip connection. This design can decrease the feature dimensions and improve the feature extraction capability. The search space of DSAS is simple and easy to understand, yet DSAS can effectively design high-performance SNN models.

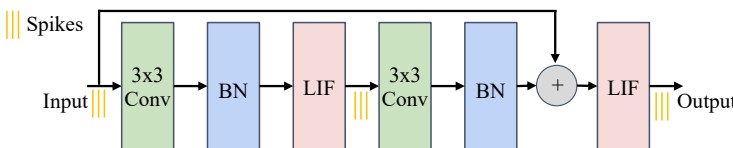

Figure 6: The normal block of the search space.

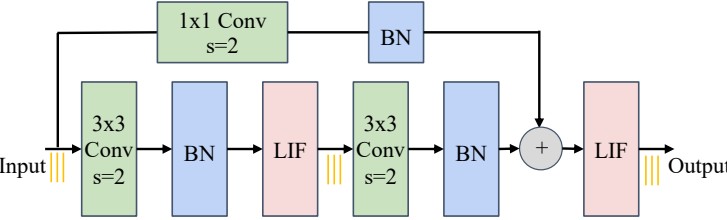

Figure 7: The downsample block of the search space.

As for the search strategy, the evolutionary search strategy designed by Sun et al. (2020) is used in the proposed DSAS method. More specifically, the process of the search strategy contains the following five steps:

1. Population Initialization: initialize the beginning generation and encode them to a sequence of real numbers;

2. Fitness Evaluation: evaluate the fitness of each individual of the population;

3. Offspring Generation: generate offspring population with crossover and mutation operators;

4. Fitness Evaluation: evaluate the fitness of each individual in the offspring population;

5. Environmental Selection: select the great individuals and put them into the next generation population.

Note that the 'Environmental Selection' step will return to Step 2 until the termination criteria are satisfied. The fitness evaluation step utilizes the proposed MAS and TGA metric, which can avoid training from scratch for each individual. This design can accelerate the search speed of the proposed DSAS method. Also, the individual cache is used to further improve the search speed as Song et al. (2025) and Sun et al. (2020). Additionally, the crossover operator is the one-bit crossover, which is commonly used in the evolutionary computation community (Sun et al., 2020). Moreover, the mutation operator has three types, i.e., adding layers, deleting layers, and randomly changing the channel of the convolution layer.

## C  THEORETICAL INSIGHTS OF DSAS

In this section, we will provide some theoretical insights to support the effectiveness of the proposed DSAS method. Specifically, we focus on the analysis of the core components of DSAS, i.e., the proposed MAS metric and TGS metric.

### C.1  THEORETICAL INSIGHTS OF MAS

The effectiveness of MAS can be analyzed from the approximate distance perspective. Specifically, the Multi-layer Activation Similarity (MAS) loss aligns the activation maps of teacher and student:

$$\mathcal{M}_T = \frac{(\bar{A}'_T) \cdot (\bar{A}'_T)^\top}{\|(\bar{A}'_T) \cdot (\bar{A}'_T)^\top\|_2}, \quad \mathcal{M}_S = \frac{(\bar{A}'_S) \cdot (\bar{A}'_S)^\top}{\|(\bar{A}'_S) \cdot (\bar{A}'_S)^\top\|_2}, \quad \mathcal{F}_{\text{MAS}} = \frac{1}{B^2} \|\mathcal{M}_T - \mathcal{M}_S\|_2^2, \quad (11)$$

where $B$ denotes the batch size, and $\bar{A}'_T$ and $\bar{A}'_S$ represent the sparsified activation maps of the ANN teacher and SNN student, respectively.

Here, the entries of $\mathcal{M}_T$ and $\mathcal{M}_S$ are given by $(\mathcal{M}_T)_{b_1 b_2} = \langle a_{b_1}^T, a_{b_2}^T \rangle$ and $(\mathcal{M}_S)_{b_1 b_2} = \langle a_{b_1}^S, a_{b_2}^S \rangle$. Consequently, $\mathcal{F}_{\text{MAS}}$ can be equivalently expressed as:

$$\mathcal{F}_{\text{MAS}} = \frac{1}{B^2} \sum_{b_1, b_2} \|\langle a_{b_1}^T, a_{b_2}^T \rangle - \langle a_{b_1}^S, a_{b_2}^S \rangle\|_2^2, \quad (12)$$

where $a_{b_1}^T$ and $a_{b_2}^T$ denote the activation vectors of the $b_1$-th and $b_2$-th samples of the ANN teacher, and $a_{b_1}^S$ and $a_{b_2}^S$ denote the corresponding student activations.

Therefore, MAS aims to minimize the discrepancy $\langle a_{b_1}^T, a_{b_2}^T \rangle - \langle a_{b_1}^S, a_{b_2}^S \rangle$, which makes $\langle a_{b_1}^T, a_{b_2}^T \rangle \approx \langle a_{b_1}^S, a_{b_2}^S \rangle$. Moreover, the activation norms of students and teachers are similar. i.e., $\|a_b^T\|^2 \approx \|a_b^S\|^2$, which is achieved by normalization and sparsity alignment.

The sample distance of ANN teacher can be defined as:

$$\|a_{b_1}^T - a_{b_2}^T\|^2 = \|a_{b_1}^T\|^2 + \|a_{b_2}^T\|^2 - 2 \cdot \langle a_{b_1}^T, a_{b_2}^T \rangle, \quad (13)$$

and similarly, the sample distance in the student space is given by:

$$\|a_{b_1}^S - a_{b_2}^S\|^2 = \|a_{b_1}^S\|^2 + \|a_{b_2}^S\|^2 - 2 \cdot \langle a_{b_1}^S, a_{b_2}^S \rangle. \quad (14)$$

As a result, the distances between any two samples in the student activation map approximately match those in the teacher:

$$\|a_{b_1}^T - a_{b_2}^T\|^2 \approx \|a_{b_1}^S - a_{b_2}^S\|^2. \quad (15)$$

This indicated that when the distances between any two samples $b_1$ and $b_2$ in the student activation space closely match those in the teacher space, the student can effectively preserve the representation ability of the teacher. Thus, the proposed MAS metric can effectively guide the search direction close to the ANN teacher and discover high-performance SNN students.

## C.2 THEORETICAL INSIGHTS OF TGS

We further analyze TGS from the perspective of gradient direction alignment. Specifically, TGS is formulated as:

$$\mathcal{P}_T = \frac{(G_T) \cdot (G_T)^\top}{\|(G_T) \cdot (G_T)^\top\|_2}, \quad \mathcal{P}_S = \frac{(G_S) \cdot (G_S)^\top}{\|(G_S) \cdot (G_S)^\top\|_2}, \quad \mathcal{F}_{\text{TGS}} = \frac{1}{N \cdot C'} \|\mathcal{P}_T - \mathcal{P}_S\|_2^2, \quad (16)$$

where $G_T$ and $G_S$ denote the gradient maps of the ANN teacher and the SNN student, respectively. The elements of $\mathcal{P}_T$ and $\mathcal{P}_S$ can be explicitly expressed as:

$$(\mathcal{P}_T)_{ij} = \frac{\langle g_i^T, g_j^T \rangle}{\|(G_T) \cdot (G_T)^\top\|_2}, \qquad (\mathcal{P}_S)_{ij} = \frac{\langle g_i^S, g_j^S \rangle}{\|(G_S) \cdot (G_S)^\top\|_2}, \quad (17)$$

where $i$ and $j$ represent the gradient vector of the $i$-th and $j$-th categories in the network.

After applying TGS, we have $(\mathcal{P}_T)_{ij} \approx (\mathcal{P}_S)_{ij}$, which implies:

$$\frac{\langle g_i^T, g_j^T \rangle}{\|(G_T) \cdot (G_T)^\top\|_2} \approx \frac{\langle g_i^S, g_j^S \rangle}{\|(G_S) \cdot (G_S)^\top\|_2}. \quad (18)$$

Thus, we can multiply both sides by the respective normalization factors and yield:

$$\langle g_i^S, g_j^S \rangle \approx \frac{\|(G_T) \cdot (G_T)^\top\|_2}{\|(G_S) \cdot (G_S)^\top\|_2} \langle g_i^T, g_j^T \rangle. \quad (19)$$

Let the global scaling constant $C = \|(G_T) \cdot (G_T)^\top\|_2 / \|(G_S) \cdot (G_S)^\top\|_2$, the equation can be simplified as:

$$\langle g_i^S, g_j^S \rangle \approx C \cdot \langle g_i^T, g_j^T \rangle. \quad (20)$$

Then, dividing both sides by the norms of the respective gradient vectors leads to:

$$\frac{\langle g_i^S, g_j^S \rangle}{\|g_i^S\| \cdot \|g_j^S\|} \approx \frac{\langle g_i^T, g_j^T \rangle}{\|g_i^T\| \cdot \|g_j^T\|}. \quad (21)$$

Consequently, according to the definition of cosine similarity, we obtain:

$$\cos \angle(g_i^S, g_j^S) \approx \cos \angle(g_i^T, g_j^T). \quad (22)$$

The close cosine relation between teacher and student gradients indicates that the updates of the student closely follow the descent directions of the teacher. As a result, the student explores the parameter space more efficiently and facilitates the discovery of high-performance solutions during the search stage.

## D THE INFLUENCE OF THE $\alpha$ VALUE

In the direct training method of SNNs, the Sigmoid function is often used as a substitute for the activation function to simulate the activation behavior of neurons because it can provide smooth and differentiable outputs, which helps the learning process of the network. More specifically, as shown in Figure 8, the black dots line indicates the Heaviside function. It can be defined in Equation (23):

$$H(x) = \begin{cases} 1, & x \geq 0 \\ 0, & x < 0 \end{cases}. \quad (23)$$

Since $H(x)$ is not differentiable at $x = 0$, traditional backpropagation cannot directly calculate the gradient. In order to achieve gradient calculation, the alternative gradient method uses a continuous smooth function to approximate $H(x)$, making backpropagation feasible.

In practice, the Sigmoid$(\cdot)$ function is widely used and has shown great performance on various tasks (Song et al., 2025; Na et al., 2022; Liu et al., 2024). The Sigmoid$(\cdot)$ function can be defined in Equation (24):

$$\sigma_\alpha(x) = \frac{1}{1 + e^{-\alpha x}}, \quad (24)$$

where $\alpha$ is the scale factor that affects the slope or steepness of the function. As indicated in Figure 8, the curve of the Sigmoid function becomes steeper with the increased $\alpha$ value. This can highly influence the derivation of the Sigmoid function.

Specifically, the derivation of the Sigmoid function can be represented as Equation (25):

$$\sigma'_\alpha(x) = \alpha \sigma_\alpha(x) \left(1 - \sigma_\alpha(x)\right). \tag{25}$$

As shown in Figure 8, the bigger the $\alpha$ value, the peak of the derivative becomes higher and narrower. Thus, we can modify the $\alpha$ value to control the shape of the derivation of the Sigmoid function. Motivated by this, we design the TGS metric to adjust the $\alpha$ value of the SNN students. By making the gradient distributions of the ANN teacher and the SNN student close, the SNN student is able to learn more knowledge from the ANN teacher and achieve better performance.

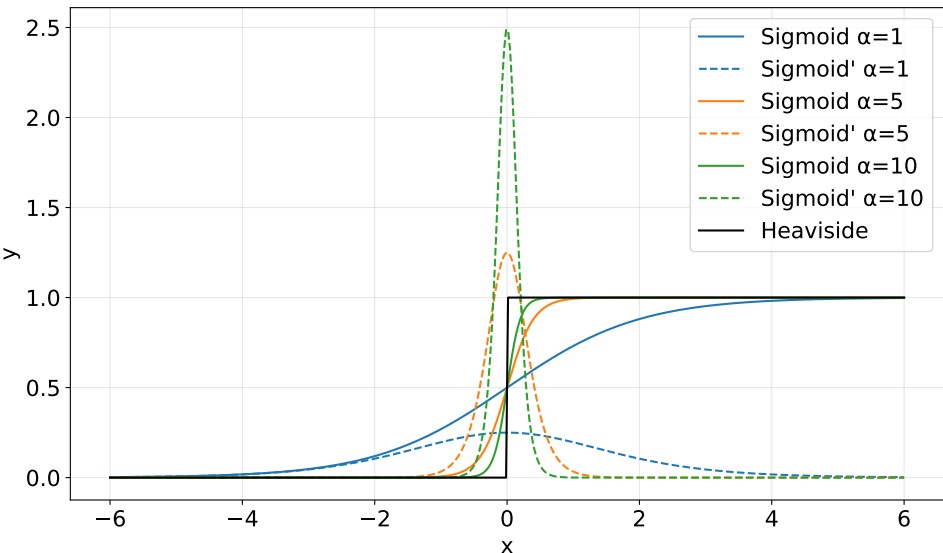

Figure 8: The Sigmoid function and its derivative with different $\alpha$ values.

## E    DETAILS OF EXPERIMENTAL SETTINGS

This is the supplementary for Section 4.1, which contains the details of the datasets, search settings, and training settings. Note that the LIF neurons with the Sigmoid surrogate gradient function and the event-to-frame integrating method for preprocessing neuromorphic datasets used in this study are implemented using SpikingJelly (Fang et al., 2023). The SGD (Robbins & Monro, 1951) optimizer is used for both static and neuromorphic datasets.

### E.1    DETAILS OF DATASETS

Four static image datasets are used to evaluate the effectiveness of the proposed DSAS method.

- **ImageNet** (Deng et al., 2009): As a milestone dataset in the field of computer vision, ImageNet contains 1.43 million high-resolution images covering 1,000 categories. These images are divided into 1.28 million training images, 50,000 validation images, and 100,000 test images. The images in this dataset are obtained from web crawling and are strictly manually annotated and verified. Mainstream research usually downsamples images to 224×224 resolution to accommodate various deep learning models.

- **Tiny-ImageNet** (Le & Yang, 2015): It is a challenging benchmark for image classification tasks. It comprises a total of 120,000 images, with 100,000 allocated for training and 10,000 for testing. These images are divided into 200 distinct categories, resulting in 500 training images and 50 testing images per category.

- **CIFAR-10** (Krizhevsky et al., 2009): It is a widely used benchmark dataset for color image classification, containing 60,000 images of 32×32 pixels, of which 50,000 are training sets and 10,000 are test sets. The dataset was manually screened from a library of 80 million micro-images, and includes categories such as airplanes, cars, birds, cats, deer, dogs, frogs, horses, ships, and trucks, with 6,000 samples in each category.

- **CIFAR-100** (Krizhevsky et al., 2009): CIFAR-100 maintains the same data scale (60,000 images) and resolution (32×32 pixels) as CIFAR-10, but expands the number of categories to 100 fine categories and introduces a hierarchical classification structure (20 superclasses combined with 100 subclasses). Since each category contains only 500 training samples, this dataset places higher demands on the small sample learning ability of deep learning models.

Two neuromorphic image datasets are chose to evaluate the performance of DSAS.

- **DIFAR10-DVS** (Li et al., 2017): It is a commonly used event stream neuromorphic dataset, containing 10,000 images, of which 9,000 are training sets and 1,000 are test sets. Each dynamic image in this dataset is obtained by converting their corresponding CIFAR-10 static images into event streams through DVS. Each dynamic image has 128×128 pixels and contains 10 types of objects such as airplanes and cars. Each type generates event sequences by 3 motion modes (translation, rotation, and scaling).

- **DVS128-Gesture** (Amir et al., 2017):This dataset uses DVS to capture the microsecond-level dynamic changes of 11 gestures (waving, clapping, etc.). Each gesture contains scene change information such as 3 types of light intensities and 4 shooting angles, and the generated image pixels are 128×128. Since this dataset contains background noise in real scenes (such as sudden changes in ambient light and limb occlusion), it can evaluate the ability of deep learning models to extract spatiotemporal invariant features.

### E.2 DETAILS OF SEARCH SETTINGS

In this section, we first provide the common settings of the search process. As presented in Table 17, the population size is set to 200, and the number of generations is 20. The genetic operation probabilities are configured as 0.9 and 0.2, i.e., 0.9 for the crossover operator and 0.2 for the mutation operator. The mutation probabilities are set to 0.7, 0.15, and 0.15 for adding layers, deleting layers, and randomly changing channel operators. Note that except for the population size setting, other settings are same as Sun et al. (2020). We utilize a larger populations to broadly exploit and explore the search space.

Table 17: The common settings of the search process.

| Population Size | Generation Size | Genetic Probability | Mutation Probability |
|:---:|:---:|:---:|:---:|
| 200 | 20 | 0.9, 0.2 | 0.7, 0.15, 0.15 |

Here we also provide the search settings of different datasets. As presented in Table 18, different datasets have different search settings on the optional convolution channel, the number range of the downsample basic block, and the number range of the normal basic block, which are denoted as 'Channel Set', 'Downsample Range', and 'Convolution Range', respectively. More specifically, the 'Channel Set' is the integer multiple or division of the last layer of channels in the ANN teacher. This is because the gradient similarity needs to be calculated with the multiple or divided ANN teacher channels. Moreover, 'Downsample Range' and 'Convolution Range' are set based on the hardware restrictions. All these values can avoid 'out of memory' on the single NVIDIA 3090 card with 24G GPU memory.

### E.3 DETAILS OF TRAINING SETTINGS

This section further details the training settings for each dataset. More specifically, for the Tiny-ImageNet dataset, the batch size is 64, the learning rate is 1e-2, the weight decay is 1e-6, and the model is trained using three NVIDIA 2080Ti GPUs. For both the CIFAR-10 and CIFAR-100

Table 18: The different search settings on various dataset.

| Dataset | Channel Set | Downsample Range | Normal Range |
|---|---|---|---|
| Tiny-ImageNet | {143, 286, 572} | {3, 3} | {3, 12} |
| CIFAR-10 & CIFAR-100 | {64, 128, 256, 512} | {2, 3} | {3, 15} |
| CIFAR10-DVS | {143, 286, 572} | {3, 3} | {3, 6} |
| DVS128-Gesture | {32, 64, 128} | {3, 3} | {3, 6} |

datasets, the batch size is 64, the learning rate is 1e-2, the weight decay is 1e-6, and the model is trained on a single NVIDIA 3090 GPU. All models are trained using 300 epochs following Guo et al. (2023). As for the neuromorphic dataset, the batch size is 32, the learning rate is 1e-2, the weight decay is 1e-6 , and the training epoch is set to 200. Note that the top-1 accuracy of the pre-trained ANN teachers, i.e., ResNet (He et al., 2016) and PyramidNet (Han et al., 2017), used in the training process is shown in Table 19.

Table 19: The top-1 accuracy of the ANN teachers used in DSAS. '-' indicate the ANN model is not selected as teacher on the corresponding dataset.

| Teacher \ Dataset | Tiny-ImageNet | CIFAR-100 | CIFAR-10 | CIFAR10-DVS | DVS128-Gesture |
|---|---|---|---|---|---|
| PyramidNet200 | 67.45 | 82.86 | 97.06 | - | - |
| PyramidNet110 | - | 80.74 | 96.17 | 82.10 | 97.92 |
| ResNet-50 | - | 78.60 | 95.01 | - | - |
| ResNet-34 | - | 78.98 | 95.12 | - | - |
| ResNet-18 | - | 75.83 | 94.13 | - | - |

# F  VISUALIZATION RESULTS OF THE SEARCHED ARCHITECTURES

In this section, we continue to present the searched architectures on various datasets. Specifically, for the static datasets, the architectures designed by the proposed DSAS method exhibit design patterns closely aligned with the complexity of the datasets and their input resolutions. As shown in Figure 9, for high-resolution ImageNet, the architecture employs a deep encoding structure comprising 12 main modules, i.e., 7 normal blocks and 5 downsampling blocks. It achieves hierarchical feature extraction through frequent fluctuations in channel numbers, e.g., 106→424→106, and multi-stage downsampling, while maintaining high channel capacity to process complex visual information. Moreover, as presented in Figure 10, the searched architecture on Tiny-ImageNet has nine modules featuring symmetrical channel changes, e.g., 286→572→286, and a moderate number of downsampling operations. When facing the simple dataset CIFAR, the searched architecture contains less basic block, as illustrated in Figure 11. We also provided the search architecture by jointly optimizing architecture and SNN neuron type, as shown in Figure 14.

In addition, the searched architectures on the neuromorphic dataset, i.e., the CIFAR10-DVS and DVS128-Gesture datasets, are shown in Figure 12 and Figure 13, respectively. The searched architecture of CIFAR10-DVS has one more normal block than the searched architecture of DVS128-Gesture. It is reasonable that CIFAR10-DVS is more complex than DVS128-Gesture.

Note that all architectures adhere to a basic pattern of alternating between normal and downsampling blocks, yet they achieve adaptive optimization on various datasets by dynamically adjusting module depth, the magnitude of channel changes, and downsampling numbers.

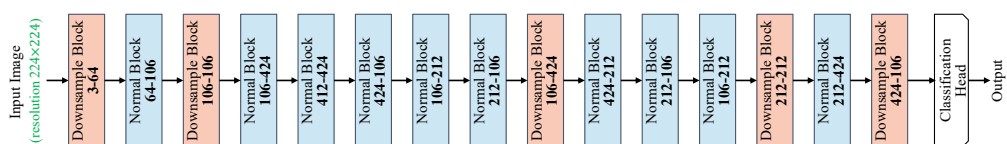

Figure 9: The searched architecture on the ImageNet dataset.

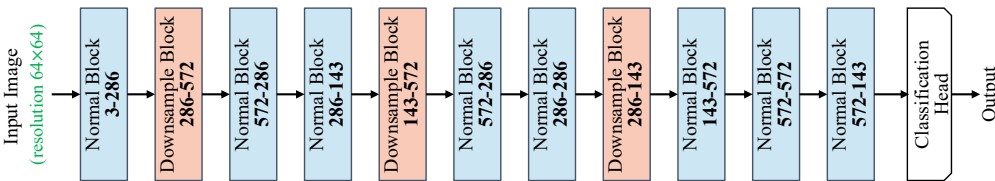

Figure 10: The searched architecture on the Tiny-ImageNet dataset.

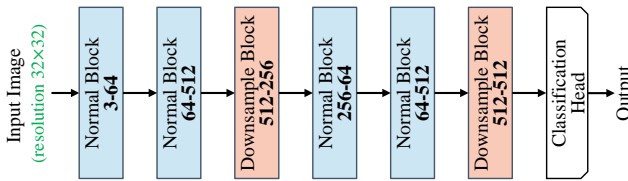

Figure 11: The searched architecture on the Tiny-ImageNet dataset.

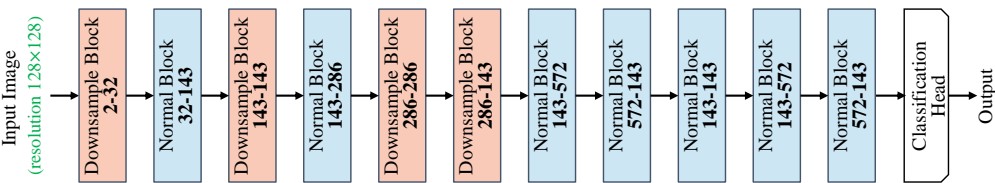

Figure 12: The searched architecture on the CIFAR10-DVS dataset.

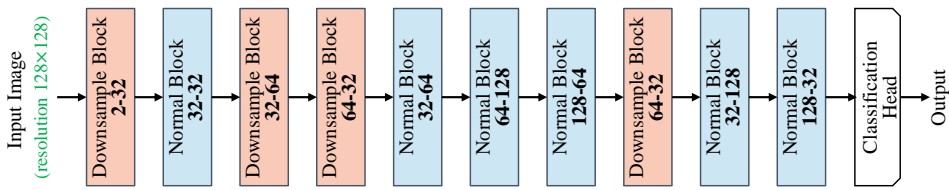

Figure 13: The searched architecture on the DVS128-Gesture dataset.

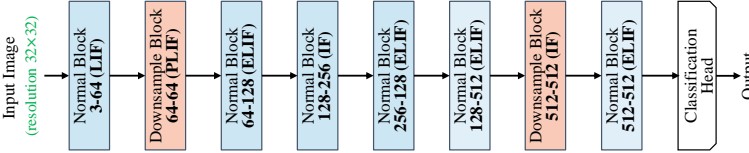

Figure 14: The searched architecture on the CIFAR dataset with the optimized neuron types.

## G    EVOLUTIONARY TRAJECTORY UNDER DIFFERENT DATASETS

This section provides the evolutionary trajectory on various datasets to justify the proposed DSAS method can successfully converge to the good fitness values, i.e., the MAS+TGS metric. Specifically, Figure 15 indicates the evolutionary trajectory on ImageNet, where the best individual is achieved at the last generation, and the evolutionary process becomes stable at the 10-th generation. Moreover, as shown in Figure 16, the best individual emerges by the 10-th generation and subsequently stabilizes on the Tiny-ImageNet dataset. Additionally, on the CIFAR dataset, the evolutionary progress is more stable than the ImageNet and Tiny-ImageNet datasets, as presented in Figure 17. Finally, as for the neuromorphic dataset, both of them converge faster than the static datasets, as shown in Figure 18 and Figure 19. This is because the scale of the static dataset is more complex than neuromorphic ones. Thus, static datasets need more generations to converge.

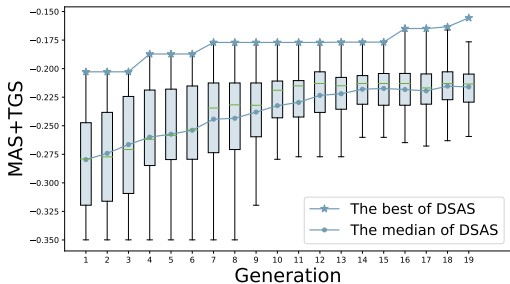

Figure 15: The evolutionary trajectory on the ImageNet dataset.

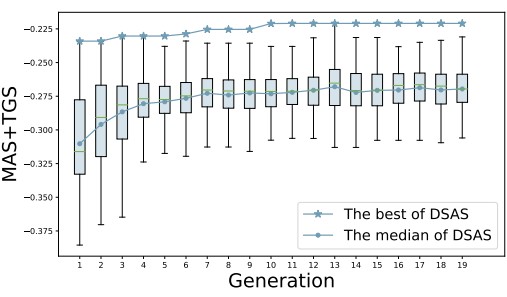

Figure 16: The evolutionary trajectory on the Tiny-ImageNet dataset.

Figure 17: The evolutionary trajectory on the CIFAR dataset.

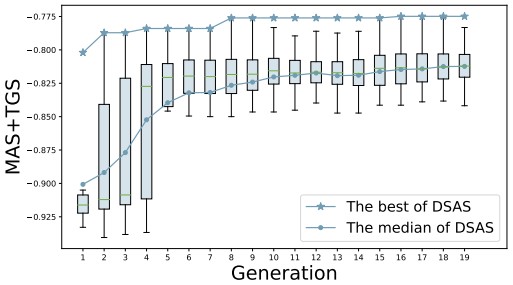

Figure 18: The evolutionary trajectory on the CIFAR10-DVS dataset.

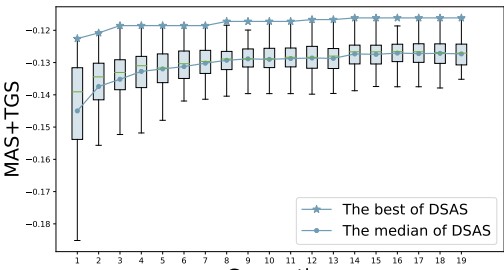

Figure 19: The evolutionary trajectory on the DVS128-Gesture dataset.

# H    THEORETICAL ENERGY CONSUMPTION ANALYSIS

The section provides the details of the theoretical energy consumption of the search model. Specifically, the energy evaluation procedure follows the standard methodology commonly adopted in the SNN community (Liu et al., 2024; Yin et al., 2021; Zhou et al., 2023). The overall process consists of three stages:

1. computing the floating-point operations (FLOPs) of each network layer;

2. converting FLOPs into spiking operations (SOPs) based on the spiking firing rate;

3. estimating the total energy consumption according to per-operation energy cost.

## H.1    STEP 1: FLOPS CALCULATION

The FLOPs of each layer are computed using Equations (26) and (27), depending on the layer type. For a convolutional layer, the FLOPs are given by:

$$\text{FLOPs}_{\text{Conv}}(l) = 2k^2 \cdot h_{out} \cdot w_{out} \cdot c_{in} \cdot c_{out}, \tag{26}$$

and for a linear layer,

$$\text{FLOPs}_{\text{Linear}}(l) = d_{\text{in}} \cdot d_{\text{out}}, \tag{27}$$

where $\text{FLOPs}_{\text{Conv}}(l)$ and $\text{FLOPs}_{\text{Linear}}(l)$ denote the floating-point operations of the $l$-th convolutional and linear layers, respectively. The variable $k$ is the kernel size; $(h_{\text{out}}, w_{\text{out}})$ represent the height and width of the output feature map; $c_{\text{in}}$ and $c_{\text{out}}$ are the input and output channel numbers; and $d_{\text{in}}$ and $d_{\text{out}}$ are the input and output dimensions of the linear layer.

## H.2    STEP 2: SOPS CONVERSION

The spiking firing rate $R$ of each layer is computed as

$$R = \frac{N_{\text{spikes}}}{N},$$

where $N_{\text{spikes}}$ denotes the number of fired spike neurons and $N$ is the total number of neurons in the layer. Based on the firing rate $R$, the SOPs of each layer are computed as:

$$\text{SOPs}(l) = T \cdot R \cdot \text{FLOPs}(l), \tag{28}$$

where $T$ is the timestep of SNNs.

## H.3    STEP 3: ENERGY CONSUMPTION ESTIMATION

The total energy consumption is obtained using Equations (29) and (30). For layers operating with floating-point MAC operations, i.e., the first layer of the searched model, the energy consumption is computed as:

$$E_F(l) = E_{\text{MAC}} \cdot \text{FLOPs}(l), \tag{29}$$

while for spiking layers, i.e., other layers except layer one, the energy consumption is computed as:

$$E_S(l) = E_{\text{AC}} \cdot \text{SOPs}(l). \tag{30}$$

Here, $E_{\text{MAC}} = 4.6\,\text{pJ}$ and $E_{\text{AC}} = 0.9\,\text{pJ}$ denote the energy consumption of a 32-bit floating-point MAC operation and an AC operation, respectively. These values follow the conventions of the SNN community (Liu et al., 2024; Yin et al., 2021; Zhou et al., 2023) and correspond to measurements obtained on a 45 nm chip.

Table 20: Performance comparison of different search & optimization methods on DVS128-Gesture.

| Optimization Method | Nature Accuracy (%) | KD Accuracy (%) | Parameter (M) |
|---|---|---|---|
| Random Search (best) | 94.10 | 95.14 | 0.62 |
| Random Search (worst) | 91.50 | 93.07 | 1.07 |
| Multi-objective EC | 89.25 | 92.18 | 0.47 |
| Zero-Cost (SAHD) | 94.79 | 95.82 | 1.16 |
| DSAS (Ours) | 95.49 | 96.53 | 0.79 |

## I MORE EFFICACY STUDIES

Table 20 summarizes the performance of various search and optimization strategies on the neuromorphic dataset. Across all methods, DSAS achieves the highest accuracy, reaching 95.49% natural accuracy and 96.53% KD accuracy, yet with the model size of 0.79 M. In addition, Zero-Cost (SAHD) attains competitive accuracy (94.79%/95.82%) but requires the largest model, i.e., 1.16 M parameters. The proposed DSAS outperforms both the best and worst random search methods in terms of accuracy and also achieves better performance compared to the multi-objective EC method.

