# OpenReview forum: "Distilling SNN Students from ANN Teachers via Spiking Neural Architecture Search"
_ICLR.cc/2026/Conference — Submitted to ICLR 2026_

### Official Review · Reviewer_8Fkb · 2025-10-31

**Soundness:** 3
**Presentation:** 2
**Contribution:** 2
**Rating:** 4
**Confidence:** 4

**Summary:**

The paper introduces DSAS, a spiking neural network (SNN) architecture search method that leverages pre-trained artificial neural networks (ANNs) to guide the design of high-performance SNNs. The paper aims to narrow the performance gap between SNNs and ANNs, especially when limited to low timesteps. The method proposes two metrics, Multi-layer Activation Similarity (MAS) and Threshold-guided Gradient Similarity (TGS), to guide the search process. The experimental results show improvements over some existing methods.

**Strengths:**

- The idea of combining ANN-guided learning with neural architecture search is interesting.
- The approach of using representation-based metrics to guide the architecture search rather than relying on iterative training reduces the computational overhead. This is a notable strength, as it significantly cuts down the search cost and improves the scalability of the method.
- The results demonstrate that DSAS performs well across multiple benchmarks, including Tiny-ImageNet and CIFAR-100, achieving competitive accuracy with fewer timesteps. The paper shows that DSAS can improve performance relative to several state-of-the-art methods, which supports the authors' claims about the method's efficacy.

**Weaknesses:**

- According to Equations 4 and 9, the alignment of activations and gradients is conceptually aimed at minimizing the angular difference, ensuring that the student SNN follows a similar trajectory to the teacher ANN. However, the paper does not adequately explore the optimization implications of this alignment. Could the authors provide further theoretical insights into how DSAS contributes to improving SNN optimization?
- Some details of the paper are difficult to follow. For instance, the meaning and use of the parameter K in TGS are unclear. Additionally, the term "KD" is not defined in the paper. Moreover, there are a few typographical errors: for example, "mateices" should be "matrices" (Line 187) and "Sigmod" should be "Sigmoid" (Line 238). A more thorough examination and clearer presentation of these details are needed.
- The paper predominantly focuses on a narrow search space that includes standard CNN components. While this choice is understandable, it restricts the exploration of architectures beyond traditional CNN-based designs. This raises the question of whether the search method demonstrates clear advantages over heuristic-based hyperparameter tuning, especially given the relatively small scale of the experiments presented.

**Questions:**

- Why was PyramidNet specifically chosen as the ANN teacher model for DSAS?

- Is it possible for DSAS to be extended for selecting the optimal type of spiking neuron for SNNs?

---

> ### Author Response · Authors · 2025-11-22
> **Response to Reviewer 8Fkb [Part 1/4]**
>
> We thank you for taking time to review the paper and for pointing out areas for additional improvement of the paper. All concerns are addressed point-by-point in the following.
>
> ## Weaknesses
>
> > **According to Equations 4 and 9, the alignment of activations and gradients is conceptually aimed at minimizing the angular difference, ensuring that the student SNN follows a similar trajectory to the teacher ANN. However, the paper does not adequately explore the optimization implications of this alignment. Could the authors provide further theoretical insights into how DSAS contributes to improving SNN optimization?**
>
> Thank you for raising this important point. To address your concern, we provided the theoretical insights of the proposed DSAS method, including the insights of MAS and TGS. The changes are added in **Appendix C** and can be summarized as follows:
>
> **Theoretical Insights of MAS:**
>
> The effectiveness of MAS can be justified by analyzing the distances between sample activations in ANN teacher and SNN student. Specifically, MAS aligns the activation maps of the ANN teacher and SNN student as follows:
>
> $\mathcal{M} _ {T} = \frac{ (\bar{A}' _ {T}) \cdot (\bar{A}' _ {T})^\top }{ \| (\bar{A}' _ {T}) \cdot (\bar{A}' _ {T})^\top \| _ 2 }, \quad \mathcal{M} _ {S} = \frac{ (\bar{A}' _ {S}) \cdot (\bar{A}' _ {S})^\top }{ \| (\bar{A}' _ {S}) \cdot (\bar{A}' _ {S})^\top \| _ 2 }, \quad \mathcal{F} _ {\text{MAS}} = \frac{1}{B^2} \left\| \mathcal{M} _ {T} - \mathcal{M} _ {S} \right\| _ 2^2,$ (1)
>
> where $B$ is the batch size, and $\bar{A}' _ {T}$ and $\bar{A}' _ {S}$ denote the sparsified activation maps of ANN teacher and SNN student.
>
> Each entry in $M _ T$ and $M _ S$ is given by the inner product between sample activations:
>
> $(M _ {T}) _ {b _ 1 b _ 2} = \langle a _ {b _ 1}^T, a _ {b _ 2}^T \rangle, (M _ {S}) _ {b _ 1 b _ 2} = \langle a _ {b _ 1}^S, a _ {b _ 2}^S \rangle.$ (2)
>
> Thus, the MAS loss can be equivalently expressed as:
>
> $F_\text{MAS} = \frac{1}{B^2} \sum_{b_1, b_2} \left\| \langle a_{b_1}^T, a_{b_2}^T\rangle - \langle a_{b_1}^S, a_{b_2}^S\rangle \right\|_2^2,$ (3)
>
> where $a_{b_1}^T$ and $a_{b_2}^T$ are the activation vectors of the $b_1$-th and $b_2$-th samples of the ANN teacher, and $a_{b_1}^S$ and $a_{b_2}^S$ denote the student activations.
>
> In the proposed DSAS method, minimizing $F_{\text{MAS}}$ encourages two aspects. (a) Alignment of pairwise inner products, i.e., $\langle a_{b_1}^T, a_{b_2}^T \rangle \approx \langle a_{b_1}^S, a_{b_2}^S \rangle$. (b) MAS encourages consistent activation structures through similarity alignment and sparsity matching, which leads to approximately comparable activation norms $|a _ b^T|^2 \approx |a _ b^S|^2$.
>
> In addition, the squared distance between two samples in teacher and student spaces can then be expressed as:
>
> $\left\| a_{b_1}^T - a_{b_2}^T \right\|^2 = \left\| a_{b_1}^T\right\|^2 + \left\| a_{b_2}^T\right\|^2 - 2\cdot \langle a_{b_1}^T, a_{b_2}^T\rangle,$ (4)
>
> $\left\| a_{b_1}^S - a_{b_2}^S \right\|^2 = \left\| a_{b_1}^S\right\|^2 + \left\| a_{b_2}^S\right\|^2 - 2\cdot \langle a_{b_1}^S, a_{b_2}^S\rangle.$ (5)
>
> Since MAS aligns both inner products and norms, the sample distances in the SNN student approximately match those in the ANN teacher:
>
> $\left\| a_{b_1}^T - a_{b_2}^T \right\|^2 \approx \left\| a_{b_1}^S - a_{b_2}^S \right\|^2.$ (6)
>
> This indicates that the SNN student can effectively preserve the representation ability of the ANN teacher. Consequently, MAS provides a reliable guidance to discover high-performance SNN students that closely follow the feature space of ANN teacher.

---

> ### Author Response · Authors · 2025-11-22
> **Response to Reviewer 8Fkb [Part 2/4]**
>
> **Theoretical Insights of TGS:**
>
> To clarify the effect of TGS, we analyze it from the perspective of gradient direction alignment. Specifically, TGS is defined as:
>
> $P _ T = \frac{(G _ {T}) \cdot  (G _ {T})^\top}{\|(G _ {T}) \cdot (G _ {T})^\top\| _ 2}, \quad
> P _ S = \frac{(G _ {S}) \cdot (G _ {S})^\top}{\|(G _ {S}) \cdot (G _ {S})^\top\| _ 2}, \quad F _ {TGS} = \frac{1}{N \cdot C'} \left\| P _ T - P _ S \right\| _ 2^2,$ (7)
>
> where $G_T$ and $G_S$ refer to the gradient maps of classification head from the ANN teacher and SNN student. Each element of the gradient map can be expressed by:
>
> $(P _ T) _ {ij} = \frac{\langle g _ i^T, g _ j^T\rangle}{\|(G _ {T}) \cdot (G _ {T})^\top\| _ 2}, \qquad
> (P _ S) _ {ij} = \frac{\langle g _ i^S, g _ j^S\rangle}{\|(G _ {S}) \cdot (G _ {S})^\top\| _ 2},$ (8)
>
> where $i$ and $j$ are the gradient vector of the $i$-th and $j$-th categories.
>
> Since TGS minimizes the discrepancy between ($P_T$) and ($P_S$), we have:
>
> $\frac{\langle g _ i^T, g _ j^T\rangle}{\|(G _ {T}) \cdot (G _ {T})^\top\| _ 2} \approx
> \frac{\langle g _ i^S, g _ j^S\rangle}{\|(G _ {S}) \cdot (G _ {S})^\top\| _ 2}.$ (9)
>
> Then, multiplying by the normalization factors gives:
>
> $\langle g _ i^S, g _ j^S\rangle \approx \frac{\|(G _ {T}) \cdot (G _ {T})^\top\| _ 2}{\|(G _ {S}) \cdot (G_{S})^\top\| _ 2} \langle g _ i^T, g _ j^T\rangle.$ (10)
>
> Let $C = \|(G _ {T}) \cdot (G _ {T})^\top\| _ 2 /\|(G _ {S}) \cdot (G _ {S})^\top\| _ 2$, then:
>
> $\langle g_i^S, g_j^S\rangle \approx C \cdot \langle g_i^T, g_j^T\rangle.$ (11)
>
> After that, normalizing both sides by the gradient norms yields:
>
> $\frac{\langle g_i^S,g_j^S\rangle}{\|g_i^S\|\cdot\|g_j^S\|} \approx \frac{\langle g_i^T,g_j^T\rangle}{\|g_i^T\|\cdot\|g_j^T\|}.$ (12)
>
> Finally, by the definition of Cosine:
>
> $\cos\angle(g_i^S,g_j^S) \approx \cos\angle(g_i^T,g_j^T).$ (13)
>
> This shows that TGS enforces the alignment between the gradient directions of the ANN teacher and SNN student. Therefore, the weight updates of SNN students follow the descent directions of the ANN teacher more closely. This design can enable a more efficient exploration of the parameter space and facilitate the discovery of high-quality architectures during the search process.
>
> > **Some details of the paper are difficult to follow. For instance, the meaning and use of the parameter K in TGS are unclear. Additionally, the term "KD" is not defined in the paper. Moreover, there are a few typographical errors: for example, "mateices" should be "matrices" (Line 187) and "Sigmod" should be "Sigmoid" (Line 238). A more thorough examination and clearer presentation of these details are needed.**
>
> We apologize for the lack of clarity in the original manuscript. We have addressed your comments as follows:
>
> 1. Clarification of the parameter $K$ in TGS: We have added a detailed explanation of the meaning and role of the parameter $K$ in **Section 3.3** (Lines 217-219).
> 2. Definition of "KD": The term "KD" is the abbreviation of "Knowledge Distillation" and has now been clearly defined at its first occurrence in the manuscript (**Section I**, Line 47).
> 3. Typos: We have corrected the typographical errors identified: "mateices" → "matrices" in Line 192 (previous Line 187) and "Sigmod" → "Sigmoid" in Line 238, 240, and 242.

---

> ### Author Response · Authors · 2025-11-22
> **Response to Reviewer 8Fkb [Part 3/4]**
>
> > **The paper predominantly focuses on a narrow search space that includes standard CNN components. While this choice is understandable, it restricts the exploration of architectures beyond traditional CNN-based designs. This raises the question of whether the search method demonstrates clear advantages over heuristic-based hyperparameter tuning, especially given the relatively small scale of the experiments presented.**
>
> Thank you for your valuable comment. We address your concern from two aspects:
>
> + **Extending our method to the Transformer-based search space.** We constructed a Transformer-based search space following Spikformer [1] designs. Specifically, we utilize the Spikformer encoder as the basic block of the search space and search the depth and hidden dimension of the model, following Auto-Spikformer [2] and AutoST [3]. As shown in the table below, DSAS achieves the optimal performance among all peer competitors. These results demonstrate that DSAS is not restricted to CNN-based search spaces and can effectively explore and optimize architectures in other search space. These experiments are added in **Section 4.4**.
> + **Comparing our search method with different optimization method.** We conducted a systematic comparison with different optimization methods, including Bayesian hyperparameter optimization [4], random search,  mulit-objective Evolutionary Computation (EC) [5], and zero-cost proxy [6]. Specifically, we use Bayesian optimization to optimize the learning rate and channel number of the optimal searched architecture. Moreover, we use random search by employing the fitness function of DSAS to justify the effectiveness of our search method. For stronger search-based optimization methods, we further include evolutionary multi-objective optimization and Zero-Cost proxy. As presented in the table below, DSAS consistently achieves the best accuracy of 81.97%. Bayesian optimization performs the weakest, and random search offers improvement but still remains inferior to DSAS. Even advanced methods such as mulit-objective EC and zero-cost proxy are surpassed by DSAS in accuracy. These results confirm that DSAS offers substantial improvements in performance compared to other optimization methods. The results and analysis are added in **Section 4.4**.
>
> **Performance comparison on the Transformer-based search space for CIFAR datasets:**
>
> |Method|Architecture|Parameter (M)|CIFAR-10 Accuracy (%)|CIFAR-100 Accuracy (%)|
> |-|-|-|-|-|
> |Spikformer|Spikformer-4-384|9.32|95.19|77.86|
> |Auto-Spikformer|Searched|4.69|95.23|77.91|
> |AutoST|Searched|11.52|96.03|79.44|
> |DSAS (Ours)| Searched|12.89|96.50|83.86|
>
> **Performance comparison of different optimization methods on CIFAR-100:**
>
> |Optimization Method|Nature Accuracy (%)|KD Accuracy (%)|Parameter (M)|
> |-|-|-|-|
> |Bayesian Optimization|53.25|60.23|16.83|
> |Random Search|73.50|80.87|10.68|
> |Mulit-objective EC|69.59|74.50|3.57|
> |Zero-Cost (SAHD)|74.7|78.86|27.18|
> |DSAS (Ours)|74.90|81.97|15.78|

---

> ### Author Response · Authors · 2025-11-22
> **Response to Reviewer 8Fkb [Part 4/4]**
>
> ## Questions
>
> > **Why was PyramidNet specifically chosen as the ANN teacher model for DSAS?**
>
> Thank you for your comment. The choice of PyramidNet is based on the following two reasons. First, we follow KDSNN [7] and choose the PyramidNet series as the ANN teacher model. Second, in our experiments, when training ANN-based ResNet models from scratch, we found that it was difficult to get very high accuracy like previous work, e.g., ResNet-18 achieving 96.30% in Joint A-SNN [8] and 94.13 in DSAS. Thus, we select the PyramidNet series with similar performance compared to the ResNet series of other methods.
>
> Actually, we have conducted experiments comparing the performance of SNN students under different teacher networks (see Table 4 in **Section 4.2**). For your convenience, the results are summarized in the table below. Specifically, across these settings, DSAS consistently produces strong SNN students with the simplest KD method, i.e., only with Logits loss. For example, the DSAS students sometimes even surpass the teacher (e.g., ResNet-32: ANN teacher 95.12% → SNN student 95.22%), yet others still have the teacher-student performance gap. This indicates that DSAS does not heavily depend on the specific teacher network and can achieve stable performance across different teacher architectures.
>
> **The performance of DSAS under the guidance of different ANN teacher on CIFAR-10:**
>
> |Method|Teacher / Accuracy (%)|Distillation Method|Student / Accuracy (%)|
> |-|-|-|-|
> |KDSNN|Pyramidnet-18 / 95.10|KL divergence loss|ResNet-18 / 93.41|
> |Joint A-SNN|ResNet-18 / 95.74|Multiple branches & KLD loss|ResNet-18 / 95.45|
> |BKDSNN|ResNet-19 / 95.60| Logits loss & Mixed distillation loss |ResNet-19 / 94.64|
> |SAKD|ResNet-19 / 96.30| Feature KD loss & Logits loss| ResNet-19 / 96.06|
> |DSAS (Ours)|ResNet-18 / 94.13|Logits loss|Searched Arch. / 94.73|
> |DSAS (Ours)|ResNet-32 / 95.12|Logits loss|Searched Arch. / 95.22|
> |DSAS (Ours)|ResNet-50 / 95.01|Logits loss|Searched Arch. / 95.18|
> |DSAS (Ours)|PyramidNet-110 / 96.17|Logits loss| Searched Arch. / 95.90|
> |DSAS (Ours)|PyramidNet-200 / 97.06|Logits loss| Searched Arch. / 96.55|
>
>
> > **Is it possible for DSAS to be extended for selecting the optimal type of spiking neuron for SNNs?**
>
> Thank you for your insightful comment. DSAS can be extended for choosing the spiking neuron types of SNNs. Specifically, DSAS has the potential to simultaneously explore the network architecture and the spiking neuron types. First, the architectures with multiple neuron-type candidates are initialized during the population initialization stage. Then, different spiking neuron types can be added in the architectures by the evolutionary operations, e.g., the crossover and mutation operations. Finally, the architecture with the great fitness value will be kept during the evolution stage.
>
> To verify the feasibility of this idea, we select four neuron types, i.e., IF [9], LIF [9], PLIF [10], and ELIF [11], and conduct the experiments on CIFAR datasets. As shown in the table below, our method still yields great performance, achieving 96.12% on CIFAR-10 and 81.08% on CIFAR-100. These results indicate that DSAS can flexibly incorporate neuron-type selection. The experiments are added in **Section 4.4**.
>
> At the same time, we would like to note that DSAS is primarily designed to focus on higher-level structural optimization, e.g., block-level selections. To improve the performance of DSAS in neuron type selection, we will focus on tailoring the neuron-level fitness function and neuron-level evolutionary method in future work.
>
> **DSAS performance when jointly optimizing architecture and neuron types on the CIFAR dataset:**
>
> |Dataset|Accuracy (%)|
> |-|-|
> |CIFAR-10|96.12|
> |CIFAR-100|81.08|
>
> **References**
>
> [1] Spikformer: When spiking neural network meets Transformer[C]. ICLR, 2023.
>
> [2] Auto-Spikformer: Spikformer architecture search[J]. Frontiers in Neuroscience, 2024.
>
> [3] AutoST: Training-free neural architecture search for spiking transformers[C]. ICASSP, 2024.
>
> [4] Optuna: A next-generation hyperparameter optimization framework[C]. ACM SIGKDD, 2019.
>
> [5] Evolutionary multi-objective spiking neural architecture search for image classification[J]. IEEE Transactions on Evolutionary Computation, 2025.
>
> [6] Neural architecture search for spiking neural networks[C]. ECCV, 2022.
>
> [7] Constructing deep spiking neural networks from artificial neural networks with knowledge distillation[C]. CVPR, 2023.
>
> [8] Joint A-SNN: Joint training of artificial and spiking neural networks via self-distillation and weight factorization[J]. Pattern Recognition, 2023.
>
> [9] Neuronal dynamics: From single neurons to networks and models of cognition[M]. Cambridge University Press, 2014.
>
> [10] Incorporating learnable membrane time constant to enhance learning of spiking neural networks[C]. ICCV, 2021.
>
> [11] Adaptive exponential integrate-and-fire model as an effective description of neuronal activity[J]. Journal of Neurophysiology, 2005.

---

> > ### Comment · Reviewer_8Fkb · 2025-11-28
> >
> > I appreciate the authors’ efforts in extending the experiments to Transformer architectures and in exploring additional spiking neuron types. However, I still have several questions and concerns:
> >
> > 1. I requested deeper theoretical insights into the proposed metrics. Unfortunately, the current explanations are still insufficient and do not meaningfully clarify the underlying rationale or theoretical contribution.
> >
> > * From the definition in Paper Eq. (4)–(5), MAS normalizes the similarity matrices rather than the activations themselves. As a result, the metric is invariant to rescaling of activations and therefore does not enforce “similar activation norms,” contrary to the authors’ claim of "(b) Similar activation norms through normalization and sparsity matching...".
> > * More broadly, the structure of $\mathcal{M}_T, \mathcal{M}_S \in \mathbb{R}^{B \times B}$ essentially mirrors the representational similarity matrices used in Representational Similarity Analysis (RSA) [1]. Given this close conceptual alignment, the purported theoretical novelty of MAS appears limited, and the primary contribution seems to lie instead in the sparse alignment between SNN and ANN rather than in the metric itself.
> > * Regarding TGS, the response does not address a fundamental issue: surrogate gradients for SNNs are known to introduce systematic approximation errors, which can degrade gradient direction alignment. Without an analysis of how TGS mitigates this effect, or whether it is robust to such errors, the claimed benefit of gradient alignment remains speculative.
> >
> > 2. It's good to see that selecting neuron types offers additional accuracy improvements. Could the authors provide the actual search results, particularly which specific spiking neuron models were selected under different conditions?
> >
> > 3. Before closing, I would again like to acknowledge the authors’ considerable effort. Overall, the paper is a relatively complete piece of work. However, I have a central concern regarding the necessity and value of applying NAS to the ANN-to-SNN distillation pipeline.
> > * The purpose of distillation is fundamentally to transfer the teacher model’s representational structure to the student model. Yet the bottlenecks in SNNs, which stem from spike-based activations and errors induced by surrogate gradients, are structural and algorithmic, not architectural. I do not see how choosing a particular CNN/Transformer variant can meaningfully resolve these issues.
> > * In addition, recent work has already scaled ANN-to-SNN distillation to a 76B-parameter foundation model [2]. At such scale, applying NAS becomes unrealistic: (1) there is no diversified population of SNN architectures with comparable parameter budgets to even form a meaningful search space; and (2) the computational cost is prohibitive. If a method cannot scale to the regime where distillation truly matters, then NAS-based ANN-to-SNN distillation loses much of its scientific and practical significance, since it will never be able to bridge the gap between SNNs and large-scale ANN models.
> >
> > [1] Kriegeskorte N, Mur M, Bandettini P A. Representational similarity analysis-connecting the branches of systems neuroscience[J]. Frontiers in systems neuroscience, 2008, 2: 249.
> >
> > [2] Pan Y, Feng Y, Zhuang J, et al. SpikingBrain: Spiking Brain-inspired Large Models[J]. arXiv preprint arXiv:2509.05276, 2025.

---

> > > ### Author Response · Authors · 2025-12-02
> > > **Response to Reviewer 8Fkb [Part 1/3]**
> > >
> > > We sincerely thank the reviewer for the detailed and constructive feedback. We have made the clarifications and hope to address your concerns.
> > >
> > > > **I requested deeper theoretical insights into the proposed metrics. Unfortunately, the current explanations are still insufficient and do not meaningfully clarify the underlying rationale or theoretical contribution.**
> > >
> > > We sincerely thank the reviewer for raising this point. We first clarify that the primary focus of our work is not to introduce a new theoretical framework but to tackle a practical and still unresolved challenge in ANN-to-SNN distillation and spiking neural architecture search. To this end, we present performance evaluations on six different datasets, seven efficacy studies examining different aspects of our method, ablation studies demonstrating the contribution of each proposed component, and visualizations that provide intuitive insights into the behavior of our study. Then, we will try to address your concerns point-to-point.
> > >
> > > - From the definition in Paper Eq. (4)–(5), MAS normalizes the similarity matrices rather than the activations themselves. As a result, the metric is invariant to rescaling of activations and therefore does not enforce “similar activation norms,” contrary to the authors’ claim of "(b) Similar activation norms through normalization and sparsity matching...".
> > >
> > > We are sorry for not clarifying this point. The phrase “normalization and sparsity matching” in item (b) does not refer to the normalization applied in Eq. (4)–(5). Instead, it refers to (i) the normalization involved in constructing the sparsity metric in Eq. (1), and (ii) the sparsity matching mechanism defined in Eq. (2) and Eq. (3) below:
> > >
> > > $\rho = \frac{1}{B N} \sum _ {b=1}^{B} \sum _ {n=1}^{N} \mathbf{1} \left[ \bar{A}' _ {S}(b,n) \ne 0 \right],$ (1)
> > >
> > > $\frac{1}{B N} \sum _ {b=1}^{B} \sum _ {n=1}^{N} \mathbf{1} \left[ A' _ {T}(b, n) > \tau \right] = \rho,$ (2)
> > >
> > > $\bar{A}' _ {T} = \max\big(A' _ {T}(b, n) - \tau, 0\big).$ (3)
> > >
> > > These operations encourage the SNN student to match the activation structure, i.e., relative magnitudes and sparsity patterns, of the ANN teacher.
> > >
> > > To address your concern, we have revised the original claim in item (b) to the following: (b) MAS encourages consistent activation structures through similarity alignment and sparsity matching, which leads to approximately comparable activation norms $|a _ b^T|^2 \approx |a _ b^S|^2$.
> > >
> > > - More broadly, the structure of $\mathcal{M} _ {T}, \mathcal{M}_{S} \in \mathbb{R}^{B\times B}$ essentially mirrors the representational similarity matrices used in Representational Similarity Analysis (RSA) (Front. Syst. Neurosci., 2008). Given this close conceptual alignment, the purported theoretical novelty of MAS appears limited, and the primary contribution seems to lie instead in the sparse alignment between SNN and ANN rather than in the metric itself.
> > >
> > > Although $\mathcal{M} _ {T}$ and $\mathcal{M} _ {S}$ resemble the representational similarity matrices used in RSA (Front. Syst. Neurosci., 2008), this type of similarity operator is not unique to RSA and has been widely employed in knowledge distillation as a standard way of capturing pairwise relational information [1-3]. Actually, we do not claim novelty in the construction of $\mathcal{M} _ {T}$ and $\mathcal{M} _ {S}$ themselves in our paper.
> > >
> > > More importantly, the core novelty of MAS does not lie in the similarity operation itself, but in how $\mathcal{M} _ {T}$ and $\mathcal{M}_{S}$ are constructed and utilized. Specifically, we introduce a sparse alignment mechanism between SNN and ANN representations, which determines which activations are compared and how they are selectively matched across heterogeneous architectures. This selective construction of the matrices is essential to the MAS metric. Thus, while the similarity computation is widely used, the innovation of MAS lies in the process by which we obtain the alignment matrices and the sparse matching framework.
> > >
> > > - Regarding TGS, the response does not address a fundamental issue: surrogate gradients for SNNs are known to introduce systematic approximation errors, which can degrade gradient direction alignment. Without an analysis of how TGS mitigates this effect, or whether it is robust to such errors, the claimed benefit of gradient alignment remains speculative.
> > >
> > > Thank you for the valuable comment. We clarify that TGS does not aim to correct the mismatch between surrogate gradients and the true underlying gradients of SNNs. Instead, TGS aligns the actual gradients produced by the selected surrogate gradient function, e.g., the Sigmoid(·) function, with the teacher ANN gradients. In other words, TGS operates after surrogate-gradient computation and acts on the resulting gradient, rather than on the surrogate function or its approximation error itself. This design ensures that TGS keeps the surrogate gradients used to train the SNN in practice consistent with the gradients of the ANN.

---

> > > ### Author Response · Authors · 2025-12-02
> > > **Response to Reviewer 8Fkb [Part 2/3]**
> > >
> > > > **It's good to see that selecting neuron types offers additional accuracy improvements. Could the authors provide the actual search results, particularly which specific spiking neuron models were selected under different conditions?**
> > >
> > > Thank you for your insightful comment. Actually, the final architectures discovered by the proposed DSAS method do not converge to one specific spiking neuron model. Instead, the search architectures typically exhibit a mixture of different neuron types across blocks (visualization at Figure 14, **Appendix F**). This is because our DSAS framework is extended to jointly search for both the network architecture and the spiking neuron types. Here, we further clarify the overall procedure and provide the detailed example:
> > >
> > > 1. **Population initialization**: During initialization, architectures are created with multiple candidate neuron types available in each layer. For example, at this stage, we may get the architectures like "LIF-LIF-LIF", "IF-IF-IF-IF", and "PLIF-PLIF". The "-" represents the connection of the blocks.
> > > 2. **Offspring Generating**: Throughout this stage, different spiking neuron types can be introduced into the architectures through crossover and mutation operations, enabling diverse neuron-type combinations to emerge. Specifically, if we employ crossover operation to "LIF-LIF-LIF" and "IF-IF-IF-IF" at block one, we can get two new architectures like "LIF-IF-IF-IF" and "IF-LIF-LIF". Furthermore, if we utilize the mutation operation to architecture "LIF-IF-IF-IF", the neuron type at the specific position would be changed to another neuron type based on the probability. e.g., we can get the mutated architecture like "LIF-IF-PIF-IF".
> > > 3. **Environmental Selection**: In each generation, individuals with higher fitness values are preserved. This process gradually guides the search toward architectures that naturally adopt hybrid neuron-type configurations rather than a specific uniform neuron model.
> > >
> > > > **Before closing, I would again like to acknowledge the authors’ considerable effort. Overall, the paper is a relatively complete piece of work. However, I have a central concern regarding the necessity and value of applying NAS to the ANN-to-SNN distillation pipeline.**
> > >
> > > We greatly appreciate your acknowledgment of our work. We would like to take this opportunity to clarify and examine the necessity and potential benefits of applying NAS in the ANN-to-SNN distillation pipeline. Next, we will address your concerns point-to-point:
> > >
> > > - The purpose of distillation is fundamentally to transfer the teacher model’s representational structure to the student model. Yet the bottlenecks in SNNs, which stem from spike-based activations and errors induced by surrogate gradients, are structural and algorithmic, not architectural. I do not see how choosing a particular CNN/Transformer variant can meaningfully resolve these issues.
> > >
> > > We fully agree that spike-based activation dynamics and surrogate-gradient errors arise from the fundamental computational principles of SNNs. We believe focusing on the structural and algorithmic directions can help the SNN student model learn more representational structure from ANN teachers. However, our study offers a different perspective by improving SNN performance from the architectural level, helping SNNs learn more effectively from ANNs. The experiments of this study justify that designing a particular SNN architecture can improve the performance learned from ANN.
> > >
> > > Actually, as shown in the table below (more details in Table 4, **Section 4.4.1** and Table 16, **Appendix A**), our experiments can demonstrate that the architecture of SNN does highly affect the accuracy after knowledge distillation. Even when the architectures of ANN and SNN are closely aligned, e.g., ANN ResNet-50 and SNN ResNet-34, there exists a significant performance drop (more than 5% in accuracy). This finding motivated us to explore the architectural-level effect of SNN students and try to find the great SNN student to narrow the performance of ANN and SNN.
> > >
> > > **The performance of different SNN architectures under the guidance of ANN teacher ('$\downarrow$' and '$\Uparrow$' denote the performance drop and improvement of SNN student, respectively) :**
> > >
> > > |ANN teacher|Teacher CIFAR-10/CIFAR-100 Acc. (%)|SNN Student|CIFAR-10 Acc. (%)|CIFAR-100 Acc. (%)|
> > > |-|-|-|-|-|
> > > |ResNet-50|95.01/78.60|ResNet-34|93.80 ($\downarrow$ 1.21)|73.47 ($\downarrow$ 5.13)|
> > > |ResNet-50|95.01/78.60|ResNet-18|94.09 ($\downarrow$ 0.92)|77.39 ($\downarrow$ 1.21)|
> > > |ResNet-50|95.01/78.60|DSAS Searched (Ours)|95.18 ($\Uparrow$ 0.17)|79.71 ($\Uparrow$ 1.11)|
> > > |ResNet-34|95.12/78.98|ResNet-18|94.55 ($\downarrow$ 0.57)|77.13 ($\downarrow$ 1.58)|
> > > |ResNet-34|95.12/78.98|DSAS Searched (Ours)|95.22 ($\Uparrow$ 0.10)|79.70 ($\Uparrow$ 0.72)|

---

> > > ### Author Response · Authors · 2025-12-02
> > > **Response to Reviewer 8Fkb [Part 3/3]**
> > >
> > > - In addition, recent work has already scaled ANN-to-SNN distillation to a 76B-parameter foundation model (SpikingBrain, arXiv, 2025). At such scale, applying NAS becomes unrealistic: (1) there is no diversified population of SNN architectures with comparable parameter budgets to even form a meaningful search space; and (2) the computational cost is prohibitive. If a method cannot scale to the regime where distillation truly matters, then NAS-based ANN-to-SNN distillation loses much of its scientific and practical significance, since it will never be able to bridge the gap between SNNs and large-scale ANN models.
> > >
> > > We thank you for this thoughtful comment. We also appreciate your pointing us to the impressive recent work (SpikingBrain, arXiv, 2025), which scales ANN-to-SNN distillation to a 76B-parameter foundation model and highlights the great potential of SNNs at foundation-model scale.  To better address your concerns, we summarize your concerns as two points:
> > >
> > > **(1) Applying NAS to LLM is unrealistic, especially for SNN**
> > >
> > > We agree that scaling NAS to spiking LLM is indeed highly challenging. The two points you mentioned, i.e., the lack of a meaningful search space for SNN and the high computational cost, are indeed key challenges for large-scale SNN NAS. However, previous studies [4-7] have shown that using NAS for LLMs is not unrealistic and can be a good way to enhance the designs and performance of LLMs. In our future work, we will explore NAS for large-scale SNNs to overcome the challenges you mentioned.
> > >
> > > **(2) NAS-based ANN-to-SNN distillation lacks significance if it cannot scale to large models**
> > >
> > > While large-scale distillation is an important motivation, we would like to highlight that ANN-to-SNN distillation also has significant scientific and practical value for small- to medium-scale models. This is because one of the original goals of both NAS and distillation is to enable efficient model deployment in resource-constrained scenarios, where SNNs can offer advantages in this scenario. Therefore, even in smaller models, NAS-based ANN-to-SNN distillation remains scientifically and practically meaningful.
> > >
> > > **References:**
> > >
> > > [1] Relational knowledge distillation[C]. CVPR, 2019.
> > >
> > > [2] DisWOT: Student architecture search for distillation without training[C]. CVPR. 2023.
> > >
> > > [3] A new similarity-based relational knowledge distillation method[C]. ICASSP, 2024.
> > >
> > > [4] Neural architecture search for parameter-efficient fine-tuning of large pre-trained language models[C]. Findings of ACL, 2023.
> > >
> > > [5] LLaMA-NAS: Efficient neural architecture search for large language models[C]. ECCV, 2024.
> > >
> > > [6] Large language model compression with neural architecture search[C]. Workshop of NeurIPS, 2024.
> > >
> > > [7] ZeroLM: Data-Free Transformer Architecture Search for Language Models[J]. arXiv, 2025.

---

### Official Review · Reviewer_8h8x · 2025-10-31

**Soundness:** 2
**Presentation:** 1
**Contribution:** 3
**Rating:** 4
**Confidence:** 5

**Summary:**

The DSAS proposed a unique combination of ANN-supervised training and architecture search. It enhances classification accuracy by employing the Mean Activation Similarity (MAS) metric to effectively align the internal feature representations between the teacher (ANN) and the student (SNN). Furthermore, DSAS introduces the Threshold Gradient Similarity (TGS) metric, which adaptively adjusts the SNN's neuron firing thresholds; this crucial step facilitates the SNN's learning process by aligning its backpropagation steps with those of the ANN.

**Strengths:**

- The proposed method, DSAS, is noteworthy as its novelty lies in the unique combination of ANN-supervised training and architecture search.
- MAS metric is effectively employed to align the internal feature representations between the teacher (ANN) and student (SNN), thereby enhancing overall classification accuracy.
- TGS metric is introduced to adaptively adjust SNN neuron firing thresholds, facilitating the SNN's learning process by aligning its backpropagation steps with those of the ANN.

**Weaknesses:**

- The MAS metric overlooks the temporal dynamics inherent to SNNs. By relying solely on the average spike rate, the temporal information is disregarded. Consequently, distinct temporal spike patterns (e.g., '0011' vs. '1100') are the same when using MAS.
- In TGS, I think comparing SNN and ANN gradients is hard because SNNs have the time dimension ($T$). As the time steps increase, the SNN's time-based gradient becomes very different from the standard ANN's gradient, making a direct comparison difficult. How can the author solve the problem?
- The experimental comparison section is deficient. Specifically, a crucial ablation study may miss: How does the performance of the proposed architecture search method compare when the search is performed within the same defined search space but without utilizing ANN pre-trained weights, instead employing an alternative search strategy?
- In Supplemental Material F, the "The evolutionary trajectory" should be the change in accuracy after retraining, rather than merely reporting the fitness function metric. Reporting the fitness metric alone does not sufficiently demonstrate the search's efficacy; only accuracy improvements can validate the search process. Furthermore, a comparison against a random choice baseline should be included.
- The quality of the writing requires substantial revision due to numerous grammatical and expression errors. Specific issues include:
    - The "DSAS" should be introduced by its full name in the Abstract, as it is the first appearance.
    - Acronyms are discouraged in Section 074 captions.
    - In Line 148, the reference to "ANN student" appears erroneous and should likely be "SNN student"?
    - In Line 353, the acronyms "C-10" and "C-100" should be explicitly defined in the figure caption.
    - In Figure 1, the legend "W: Weight" should be corrected to "W: Width," and the legend's overall descriptive detail is insufficient.

**Questions:**

See weakness

---

> ### Author Response · Authors · 2025-11-22
> **Response to Reviewer 8h8x [Part 1/2]**
>
> We would like to thank you for your well-throughout comments and suggestions. These comments are highly important for us to improve our study. All comments are addressed point-by-point to our best knowledge.
>
> > **The MAS metric overlooks the temporal dynamics inherent to SNNs. By relying solely on the average spike rate, the temporal information is disregarded. Consequently, distinct temporal spike patterns (e.g., '0011' vs. '1100') are the same when using MAS.**
>
> Thank you for your valuable comment. The spike encoding strategy of the proposed MAS metric is closely aligned with rate-based coding in SNNs, i.e., the information is represented through the overall firing rate rather than precise spike timing. Under this coding scheme, sequences like "0011" and "1100" have the same overall intensity and are thus encoded equivalently. Rate coding remains widely used due to its simplicity, robustness [1], and hardware efficiency [2]. Therefore, MAS is designed to follow this rate-based representation, which treats spike sequences with the same firing rate as equivalent.
>
> We also recognize that temporal coding is an important aspect of SNNs, and we will explore extensions of our framework to incorporate temporal dynamics in future work.
>
> > **In TGS, I think comparing SNN and ANN gradients is hard because SNNs have the time dimension ($T$). As the time steps increase, the SNN's time-based gradient becomes very different from the standard ANN's gradient, making a direct comparison difficult. How can the author solve the problem?**
>
> We thank the reviewer for the comment. In our implementation, TGS uses gradients of the classification head. Although the SNN features have a temporal dimension $T$ during forward propagation, we average the features across time before feeding them into the classification head following conversions [3]-[5]. Thus, the resulting gradient is a single vector that aggregates temporal information. This allows appropriately comparison and alignment with the ANN classification head gradient.
>
> > **The experimental comparison section is deficient. Specifically, a crucial ablation study may miss: How does the performance of the proposed architecture search method compare when the search is performed within the same defined search space but without utilizing ANN pre-trained weights, instead employing an alternative search strategy?**
>
> Thank you for your insightful comment. To address this concern, we conduct additional experiments comparing different search strategies within the same search space but without using ANN pre-trained weights. Specifically, as presented in the table below, on the CIFAR-100 dataset, multi-objective Evolutionary Computation (EC) [5] directly uses accuracy as the fitness value, resulting in a small model but lower accuracy compared to DSAS. Moreover, the architecture guided by the zero-cost SAHD proxy [6] has a larger parameter size (27.18 M) but achieves inferior accuracy (78.86%) compared to DSAS. On the DVS128-Gesture dataset, DSAS also gets the best performance compared to other methods without pre-trained ANN weights. These results demonstrate that DSAS outperforms alternative strategies, indicating the effectiveness of the ANN-guided search method of the proposed DSAS. These results are added in **Section 4.4** and **Appendix I**.
>
> **Performance of different search strategies without pre-trained ANN weights on CIFAR-100:**
>
> | Search Strategy    | Nature Accuracy (%) | KD Accuracy  (%) | Parameter (M) |
> | ------------------ | ------------------- | ---------------- | ------------- |
> | Mulit-objective EC | 69.59               | 74.50            | 3.57          |
> | Zero-Cost (SAHD)   | 74.7                | 78.86            | 27.18         |
> | DSAS (Ours)        | 74.90               | 81.97            | 15.78         |
>
> **Performance of different search strategies without pre-trained ANN weights on DVS128-Gesture:**
>
> | Search Strategy | Nature Accuracy (%) | KD Accuracy (%) | Parameter (M) |
> |------------------------|-------------|---------|-----------|
> | Mulit-objective EC |    89.25    |    92.18     |     0.47      |
> | Zero-Cost (SAHD)  |      94.79      |    95.82     |  1.16    |
> | DSAS (Ours)            |      95.49       |    96.53     |    0.79      |

---

> ### Author Response · Authors · 2025-11-22
> **Response to Reviewer 8h8x [Part 2/2]**
>
> > **In Supplemental Material F, the "The evolutionary trajectory" should be the change in accuracy after retraining, rather than merely reporting the fitness function metric. Reporting the fitness metric alone does not sufficiently demonstrate the search's efficacy; only accuracy improvements can validate the search process. Furthermore, a comparison against a random choice baseline should be included.**
>
> Thank you for your constructive comment. Actually, visualizing the evolutionary trajectory of the real accuracy of the search architectures is computationally infeasible due to the extremely high cost. For example, training a single architecture on CIFAR-10 requires 0.25 average GPU Days, and DSAS explores 4000 candidate architectures, which would result in a total cost of $0.25×4000=1000$ GPU Days. Such an enormous computational cost makes exhaustive retraining impractical, especially considering the large search space and datasets. To address this, the fitness function metric, i.e., the proposed MAS and TGS metrics, is designed to accelerate the search process while still providing a meaningful reflection of accuracy improvements. This allows the search to efficiently identify promising architectures without incurring prohibitive computational cost.
>
> To address the concern, we add a comparison with a random search baseline on CIFAR-100 and DVS128-Gesture, combined with the proposed DSAS method. As shown in the table below, DSAS achieves the highest accuracy (74.90% Nature Acc., 81.97% KD Acc.) with a moderate model size (15.78M) compared with random search. These analyses are added in **Section 4.4** and **Appendix I**.
>
> **Performance comparison with random search baseline on CIFAR-100:**
>
> |Search Stategy|Nature Accuracy (%)|KD Accuracy (%)|Parameter (M)|
> |-|-|-|-|
> |Random Search (best)|73.50|80.87|10.68|
> |Random Search (worst)|71.40|77.91|26.72|
> |DSAS (Ours)|74.90|81.97|15.78|
>
> **Performance comparison with random search baseline on DVS128-Gesture:**
>
> |Search Stategy|Nature Accuracy (%)|KD Accuracy (%)|Parameter (M)|
> |-|-|-|-|
> |Random Search (best)|94.10|95.14|0.62|
> |Random Search (worst)|91.50|93.07|1.07|
> |DSAS (Ours)|95.49|96.53|0.79|
>
> > **The quality of the writing requires substantial revision due to numerous grammatical and expression errors. Specific issues include:**
> >
> > - The "DSAS" should be introduced by its full name in the Abstract, as it is the first appearance.
> > - Acronyms are discouraged in Section 074 captions.
> > - In Line 148, the reference to "ANN student" appears erroneous and should likely be "SNN student"?
> > - In Line 353, the acronyms "C-10" and "C-100" should be explicitly defined in the figure caption.
> > - In Figure 1, the legend "W: Weight" should be corrected to "W: Width," and the legend's overall descriptive detail is insufficient.
>
> Thank you for your meticulous review, and we are sorry for the expression errors in the previous version. We have carefully revised the manuscript to address all the concerns raised. The detailed changes are as follows:
>
> 1. Introduction of "DSAS" in the Abstract: "DSAS" is now introduced by its full name at its first appearance in the **Abstract**.
> 2. Acronym usage in Section 074 captions: All acronyms in the Section 2.1 (074), 2.2, 3.2, and 3.3 captions have been replaced with full expressions.
> 3. Correction of ''ANN student'' (Line 148): “ANN student” has been corrected to "SNN student" (see Line 154 in the revision).
> 4. Acronyms of ''C-10'' and ''C-100'' (Line 353): We have provided the detailed explanation of the acronyms utilized in the experiment section (see **Section 4.1** in Lines 268-269) for clarity. Specifically, we utilize "C-10", "C-100", "C10-DVS", and "Gesture" to denote the acronyms of the CIFAR-10, CIFAR-100, CIFAR10-DVS, and DVS128-Gesture datasets. The ResNet and PyramidNet series are denoted by "R-" and "P-", respectively.
> 5. The legend of Figure 1: We have corrected the legend to "W: Width" and expanded the details to improve clarity. Specifically, we add the legend of "ANN layer", "SNN student", "Activation map", and "Gradient map". Now, all components of Figure 1 are defined in the legend.
>
> **References**
>
> [1] Rate coding or direct coding: Which one is better for accurate, robust, and energy-efficient spiking neural networks?[C]. ICASSP, 2022.
>
> [2] Neuromorphic silicon neuron circuits[J]. Frontiers in Neuroscience, 2011.
>
> [3] Spikformer: When spiking neural network meets Transformer[C]. ICLR, 2023.
>
> [4] BKDSNN: Enhancing the performance of learning-based spiking neural networks training with blurred knowledge distillation[C]. ECCV, 2024.
>
> [5] Evolutionary multi-objective spiking neural architecture search for image classification[J]. IEEE Transactions on Evolutionary Computation, 2025.
>
> [6] Neural architecture search for spiking neural networks[C]. ECCV, 2022.

---

### Official Review · Reviewer_GNxc · 2025-11-01

**Soundness:** 3
**Presentation:** 3
**Contribution:** 3
**Rating:** 4
**Confidence:** 4

**Summary:**

This paper proposes DSAS, a novel method for designing efficient Spiking Neural Networks by integrating knowledge distillation with neural architecture search. Its core innovation lies in introducing a Multi-layer Activation Similarity metric, which guides the search process by measuring the activation similarity between the SNN student and the ANN teacher across multiple intermediate layers. The authors conduct experiments on multiple datasets including Tiny-ImageNet, ImageNet, and CIFAR, demonstrating that DSAS significantly outperforms existing methods in terms of accuracy and parameter efficiency, with low search cost.

**Strengths:**

1.The paper accurately identifies the shallow problem in existing SNAS methods, where search algorithms tend to favor shallow networks to quickly match the final output of the teacher model.

2.MAS is the central contribution of this paper. By extending alignment from the final output to multi-layer intermediate features, this metric cleverly guides the search direction.

**Weaknesses:**

1.The experiments are conducted solely on traditional CNNs. Currently, Transformer architectures have become mainstream in fields such as computer vision. To demonstrate the generalizability and state-of-the-art relevance of the proposed method, the authors should include experimental results on converting trained Transformer models from ANN to SNN.

2.In the ImageNet comparison, DSAS uses PyramidNet101 as its teacher model, while many baseline methods use ResNet-34 or ResNet-50. Using a more powerful teacher model may give DSAS an unfair advantage. It is unclear how much of the performance gain is attributable to the superior search method itself versus the stronger teacher, requiring a more rigorous analysis. And how would the performance of DSAS change if its teacher model were replaced with ResNet-34, the same as used by baseline methods.

**Questions:**

As mentioned in weaknesses

---

> ### Author Response · Authors · 2025-11-22
> **Response to Reviewer GNxc**
>
> We sincerely thank you for the time and effort you have invested in reviewing our paper. Your comments and questions are very insightful and highly valuable for us to enhance the paper. Your concerns are addressed point-by-point in the following.
>
> > **1.The experiments are conducted solely on traditional CNNs. Currently, Transformer architectures have become mainstream in fields such as computer vision. To demonstrate the generalizability and state-of-the-art relevance of the proposed method, the authors should include experimental results on converting trained Transformer models from ANN to SNN.**
>
> Thank you for your insightful comment. We address your concern by adding the experiments on the Transformer-based search space. Specifically, we adopt the encoder block of Spikformer [1] as the basic search block and follow the search settings of Auto-Spikformer [2] and AutoST [3]. As shown in the table below, the proposed DSAS method successfully searches high-performance Transformer-based SNN architectures and achieves 96.50% on CIFAR-10 and 83.86% on CIFAR-100. This indicates that DSAS outperforms existing methods, especially on the complex CIFAR-100. The experiments and discussions are added in **Section 4.4**.
>
> **Performance comparison on the Transformer-based search space for CIFAR datasets:**
>
> |Method|Architecture|Parameter (M)|CIFAR-10 Accuracy (%)|CIFAR-100 Accuracy (%)|
> |-|-|-|-|-|
> |Spikformer|Spikformer-4-384|9.32|95.19|77.86|
> |Auto-Spikformer|Searched|4.69|95.23|77.91|
> |AutoST|Searched|11.52|96.03|79.44|
> |DSAS (Ours)|Searched|12.89|96.50|83.86|
>
> > **2.In the ImageNet comparison, DSAS uses PyramidNet101 as its teacher model, while many baseline methods use ResNet-34 or ResNet-50. Using a more powerful teacher model may give DSAS an unfair advantage. It is unclear how much of the performance gain is attributable to the superior search method itself versus the stronger teacher, requiring a more rigorous analysis. And how would the performance of DSAS change if its teacher model were replaced with ResNet-34, the same as used by baseline methods.**
>
> Thank you for your valuable comment.  Our choice was motivated by two factors: (1) following KDSNN [4], which also adopts the PyramidNet family as ANN teachers, and (2) our re-implementations showed that ResNet teachers trained from scratch did not reach the high accuracies reported in prior work. We therefore chose PyramidNet teachers, whose performance is more consistent in related SNN studies.
>
> In addition, we note that most existing methods do not report the accuracy of the ANN teacher, making it difficult to perform a direct, fair, and comprehensive teacher-capacity analysis in the ImageNet setting. Therefore, we provide the experiments on CIFAR-10 to examine the influence of different ANN teachers, including ResNet-34. As shown in the table below, across all settings of ANN teachers, DSAS consistently produces strong SNN students using only Logits loss for distillation. In some cases, the DSAS student even slightly surpasses its teacher (e.g., ResNet-32: 95.12% → 95.22%), and when higher-performance teachers are used, the accuracy drop remains similarly small (e.g., 96.17% → 95.90%). These results demonstrate that DSAS is not tied to a specific teacher architecture, and its performance gains primarily originate from the architecture search itself rather than from the strength of the teacher model.
>
> **The performance of DSAS under the guidance of different ANN teacher on CIFAR-10:**
>
> |Method|Teacher / Accuracy (%)|Distillation Method|Student / Accuracy (%)|
> |-|-|-|-|
> |KDSNN|Pyramidnet-18 /  95.10|KL divergence loss|ResNet-18 / 93.41|
> |Joint A-SNN|ResNet-18 / 95.74|Multiple branches & KLD loss|ResNet-18 / 95.45|
> |BKDSNN|ResNet-19 / 95.60|Logits loss & Mixed distillation loss|ResNet-19 / 94.64|
> |SAKD|ResNet-19 / 96.30|Feature KD loss & Logits loss|ResNet-19 / 96.06|
> |DSAS (Ours)|ResNet-18 / 94.13|Logits loss|Searched Arch. / 94.73|
> |DSAS (Ours)|ResNet-32 / 95.12|Logits loss|Searched Arch. / 95.22|
> |DSAS (Ours)|ResNet-50 / 95.01|Logits loss|Searched Arch. / 95.18|
> |DSAS (Ours)|PyramidNet-110 / 96.17|Logits loss|Searched Arch. / 95.90|
> |DSAS (Ours)|PyramidNet-200 / 97.06|Logits loss|Searched Arch. / 96.55|
>
> **Reference**
>
> [1] Spikformer: When spiking neural network meets Transformer[C]. ICLR, 2023.
>
> [2] Auto-Spikformer: Spikformer architecture search[J]. Frontiers in Neuroscience, 2024.
>
> [3] AutoST: Training-free neural architecture search for spiking transformers[C]. ICASSP, 2024.
>
> [4] Constructing deep spiking neural networks from artificial neural networks with knowledge distillation[C]. CVPR, 2023.

---

### Official Review · Reviewer_5xcB · 2025-11-02

**Soundness:** 3
**Presentation:** 3
**Contribution:** 3
**Rating:** 4
**Confidence:** 3

**Summary:**

This paper proposes a novel training-free Spiking Neural Architecture Search (SNAS) method, named DSAS. Its core objective is to leverage the knowledge of pre-trained Artificial Neural Network (ANN) teachers to automatically discover high-performance and energy-efficient Spiking Neural Network (SNN) student models under low timestep conditions. The paper includes two main metrics: Multi-layer Activation Similarity (MAS) and Threshold-guided Gradient Similarity (TGS). The experimental section comprehensively covers four static image datasets (ImageNet, Tiny-ImageNet, CIFAR-10, CIFAR-100) and two neuromorphic datasets (CIFAR10-DVS, DVS128-Gesture), with systematic comparisons against various existing distillation and architecture search methods.

**Strengths:**

The authors conducted extensive experiments across six datasets (with both static and neuromorphic tasks). The evaluation includes diverse aspects of the architecture and model.

The paper's motivation is clear. The core problem is sharply focused on "how to design efficient SNN architectures that are both high-performing and require few timesteps." The approach of fusing ANN teacher knowledge with automated architecture search provides a solid and reasonable foundation for the research direction.

**Weaknesses:**

The main concern is that, the use of knowledge distillation (KD) for student network search is not new. The idea of MAS appears quite similar to Relation Similarity Metric in Dong et al. (2023), essentially extending an ANN-based method to the SNN domain. The authors should clarify the key differences between the two activation-similarity mechanisms and highlight the unique challenges, if any, in applying this approach to SNNs.

I recommend reporting the total network design time. Since the pipeline involves both the search stage and the teacher model pretraining, the overall cost should be included to provide a fair comparison.

The methodology for energy evaluation is unclear. Please describe how the energy consumption is calculated to allow the reader to better understand and reproduce the results.

**Questions:**

See the weakness.

---

> ### Author Response · Authors · 2025-11-22
> **Response to Reviewer 5xcB [Part 1/2]**
>
> We are immensely grateful for your effort in reviewing our work, as well as for your constructive suggestions and insightful questions. In the following, we will address each of your suggestions and concerns point-by-point.
>
> > **The main concern is that, the use of knowledge distillation (KD) for student network search is not new. The idea of MAS appears quite similar to Relation Similarity Metric in Dong et al. (2023), essentially extending an ANN-based method to the SNN domain. The authors should clarify the key differences between the two activation-similarity mechanisms and highlight the unique challenges, if any, in applying this approach to SNNs.**
>
> Thank you for your insightful comment. The key differences between the proposed MAS metric and the Relation Similarity Metric (RSM) can be summarized as two aspects:
>
> - **MAS can handle the temporal information unique to SNN:** The previous RSM method focuses only on spatial activation similarity in ANNs and cannot directly handle the temporal information of SNNs. In contrast, MAS explicitly accumulates student SNN activations over the temporal domain to account for spike-based temporal behavior like rate-based encoding in SNNs. This design is essential for achieving accuracy in teacher-student alignment in the SNN domain.
> - **MAS can avoid the bias of RSM toward shallow networks on SNN:** The previous RSM method considers only the last-layer activation similarity, which can bias selection toward very shallow networks. i.e.,1-2 layers, in SNNs (see Figure 4 and Figure 5). This is because shallow layers of SNN have dense activations, which are more similar to the dense activations of the ANN's last layer. Thus, previous RSM may mislead the search direction and generate inferior architecture. In comparison, MAS aligns sparsity between teacher and student and computes similarity across multiple layers. These designs can ensure fair evaluation and improve high-performance student selection.
>
> In summary, MAS not only adapts previous RSM in SNNs but also introduces a multi-layer sparsity-aware mechanism. It is crucial to search for high-performance SNN architecture in practice.
>
> > **I recommend reporting the total network design time. Since the pipeline involves both the search stage and the teacher model pretraining, the overall cost should be included to provide a fair comparison.**
>
> Thank you for your valuable suggestion. We agree that reporting the total network design time, including both the architecture search stage and the teacher model pretraining, provides a more comprehensive comparison. In the revised manuscript, we have added the overall design cost for the proposed DSAS method. As shown in the table below, DSAS\* includes the ANN teacher pretraining time in addition to the search cost. The total GPU-day cost of DSAS\* remains lower than or comparable to existing SNAS methods on both static and neuromorphic datasets. It is also worth noting that although the ImageNet cost increases when ANN teacher training is included, the pretrained weights are publicly available and easily reusable from GitHub, meaning that this cost typically does not need to be paid again in practice. These modifications are placed in **Section 4.4**.
>
> **The search cost (GPU days) of different SNAS methods ('\*' indicates that ANN teacher training time is considered.):**
>
> | Method        | ImageNet | Tiny-ImageNet | CIFAR | C10-DVS | Gesture |
> | ------------- | -------- | ------------- | ----- | ------- | ------- |
> | AutoSNN       | -        | 5.2           | 1.9   | 1.3     | -       |
> | SpikeDHS      | -        | -             | 1.4   | -       | -       |
> | LitE-SNN      | -        | -             | 5.1   | -       | -       |
> | ESNNs         | -        | 0.207         | 0.07  | 0.15    | -       |
> | SNASNet       | -        | 0.17          | 0.127 | -       | -       |
> | EMO-SNAS      | -        | 57            | 25.5  | -       | 2       |
> | DSAS (Ours)   | 0.031    | 0.044         | 0.088 | 0.052   | 0.005   |
> | DSAS\* (Ours) | 7.351    | 0.478         | 0.471 | 0.422   | 0.189   |

---

> ### Author Response · Authors · 2025-11-22
> **Response to Reviewer 5xcB [Part 2/2]**
>
> > **The methodology for energy evaluation is unclear. Please describe how the energy consumption is calculated to allow the reader to better understand and reproduce the results.**
>
> We are sorry for not clarifying this point in the initial version. In this revision, we have provided the detailed explanations of energy consumption and added them to **Appendix H**. Specifically, the energy evaluation method follows the conventions of the SNN community [1]-[3], which can be divided into three steps:
>
> - *Step 1*: Calculate the floating point operations (FLOPs) of each layer of the network.
> - *Step 2*: Convert FLOPs to the spiking operations (SOPs) based on the spiking fire rate.
> - *Step 3*: Get the total energy consumption according to the energy consumption per operation.
>
> First, *Step 1* can be calculated by Equation (1) and (2). If the layer type is a convolutional layer, the FLOPs can be evaluated using Equation (1). If the layer type is linear layer, the FLOPs can be attained using Equation (2).
>
> $\text{FLOPs} _ {\text{Conv}}(l) = 2k^2 \cdot h _ {out} \cdot w _ {out} \cdot c _ {in} \cdot c _ {out},$ (1)
>
> $\text{FLOPs} _ {\text{Linear}}(l) = d _ {in} \cdot d _ {out},$ (2)
>
> where $\text{FLOPs} _ {\text{Conv}}(l)$ and $\text{FLOPs} _ {\text{Linear}}$ are the floating-point operations of the $l$-th convolutional layer and linear layer, k is the kernel size, $(h_{out}, w_{out})$ is the height and width of the feature map, $c_{in}$ and $c_{out}$ are the input and output channels, $d_{in}$ and  $d_{out}$ are the input and output dimensions of linear layer.
>
> Then, *Step 2* is processed. Specifically, we can statistically determine the spiking fire rate $R$ of each layer using the number of fired neurons $N_{spikes}$ and the total number of neurons $N$, where $R=N_{spikes}/N$ . After that, SOPs can be calculated using Equation (3):
>
> $\text{SOPs}(l) = T \cdot R \cdot \text{FLOPs}(l),$ (3)
>
> where $\text{SOPs}(l)$ is the spiking operations of the $i$-th layer, $\text{FLOPs}(l)$ is the floating-point operations $l$-th layer according to the specific layer type, $T$ is the timestep of SNN.
>
> Finally, we can get the energy consumption of the neural network model by Equations (4) and (5). Specifically, if the floating-point operation is employed in the layer, i.e., the first layer of the search SNN model, the energy consumption can be calculated using Equation (4). Otherwise, the energy consumption of spiking layers, i.e., other layers after the first layer, can be calculated by Equation (5).
>
> $E _ {F}=E _ \text{MAC} \cdot \text{FLOPs} _ C(l)$ (4)
>
> $E _ {S}=E _ \text{AC} \cdot \text{SOPs} _ C(l)$ (5)
>
> where $E _ \text{MAC} = 4.6\,{pJ}$ and $E _ \text{AC} = 0.9\,{pJ}$ are the energy consumption of a floating-point MAC operation and an accumulate AC operation, respectively. Note that $E _ \text{MAC}$ and $E _ \text{AC}$ are calculated using 32-bit floating-point calculations on a 45 nm chip, following the conventions [1]-[3].
>
> **References**
>
> [1] Accurate and efficient time-domain classification with adaptive spiking recurrent neural networks[J]. Nature Machine Intelligence, 2021.
>
> [2] LitE-SNN: designing lightweight and efficient spiking neural network through spatial-temporal compressive network search and joint optimization[C]. IJCAI, 2024.
>
> [3] Spikformer: When spiking neural network meets Transformer, ICLR, 2023.

---

### Official Review · Reviewer_yxtV · 2025-11-02

**Soundness:** 3
**Presentation:** 3
**Contribution:** 3
**Rating:** 6
**Confidence:** 4

**Summary:**

This paper presents a training-free neural architecture search (NAS) method, named DSAS, for distilling a high-performance SNN from a pre-trained ANN. DSAS is an evolutionary NAS approach that introduces a multi-layer activation similarity (MAS) metric to align features between the ANN teacher and the SNN student, and a threshold-guided gradient similarity (TGS) method to enhance the SNN’s approximation of ANN gradients. Experimental results show that SNNs obtained by DSAS achieve competitive performance compared with state-of-the-art SNNs.

**Strengths:**

- The proposed search method relies on evaluation-based algorithms, resulting in substantially lower search costs compared to existing spiking NAS approaches.

- The paper provides a comprehensive empirical evaluation, including ablation studies, parameter analyses, and comparisons with multiple baselines.

**Weaknesses:**

- The search framework is primarily designed for spiking CNNs, without consideration of transformer-based architectures. Moreover, the SNN candidates in the search space are identical to those in EMO-SNAS [Song, TEVC 2025], which limits the novelty of the architectural design.

- DSAS performs architecture searches separately for each dataset and achieves competitive results; however, the paper lacks analysis on cross-dataset generalization — for instance, whether an SNN searched on ImageNet could transfer effectively to CIFAR10.

- Since the parameter spaces of ANNs and SNNs differ, it remains unclear how the TGS metric can meaningfully capture gradient alignment between teacher and student networks.

- Some implementation details are missing, such as the surrogate gradient function used for training the searched SNN, and the event data representations for CIFAR10-DVS and DVS128-Gesture (e.g., whether voxel grids are used).

- Typo: Line 240 — “SNN Teacher” should be “SNN Student.

**Questions:**

Please refer to the points raised in “Weaknesses”.

---

> ### Author Response · Authors · 2025-11-22
> **Response to Reviewer yxtV [Part 1/2]**
>
> Thank you sincerely for the encouragement and the recognition in our work. We appreciate your great effort and address your comments in detail below.
>
> > **The search framework is primarily designed for spiking CNNs, without consideration of transformer-based architectures. Moreover, the SNN candidates in the search space are identical to those in EMO-SNAS [Song, TEVC 2025], which limits the novelty of the architectural design.**
>
> Thank you very much for your insightful comments. To address your concern, we extend the proposed DSAS method to the Transformer-based search space by adopting the Spikformer [1] encoder block as the basic block and following the search settings of Auto-Spikformer [2] and AutoST [3]. As shown in the table below, SAS achieves 96.50% on CIFAR-10 and 83.86% on CIFAR-100, outperforming existing methods, especially on the complex CIFAR-100. This confirms that DSAS can design the architectures beyond spiking CNNs. The experiments are added in **Section 4.4**.
>
> **Performance comparison on the Transformer-based search space for CIFAR datasets:**
>
> | Method          | Architecture     | Parameter (M) | CIFAR-10 Accuracy (%) | CIFAR-100 Accuracy (%) |
> | --------------- | ---------------- | ------------- | --------------------- | ---------------------- |
> | Spikformer      | Spikformer-4-384 | 9.32          | 95.19                 | 77.86                  |
> | Auto-Spikformer | Searched         | 4.69          | 95.23                 | 77.91                  |
> | AutoST          | Searched         | 11.52         | 96.03                 | 79.44                  |
> | DSAS (Ours)     | Searched         | 12.89         | 96.50                 | 83.86                  |
>
> > **DSAS performs architecture searches separately for each dataset and achieves competitive results; however, the paper lacks analysis on cross-dataset generalization — for instance, whether an SNN searched on ImageNet could transfer effectively to CIFAR10.**
>
> Thank you for your valuable comment. We add an analysis of the transferability of searched architectures across datasets to address your concern. Specifically, we evaluate architectures searched on one dataset and directly apply them to another dataset with only modifications to the classification head. As shown in the table below, DSAS-searched architectures transfer well across datasets: Tiny-ImageNet → CIFAR-10 achieves 96.38%, ImageNet → CIFAR-10 achieves 94.02%, CIFAR-10 → Tiny-ImageNet achieves 65.41%, and CIFAR10-DVS → CIFAR-10 / DVS128-Gesture achieves 92.97% / 95.83%. These results demonstrate that the searched architectures show comparable cross-dataset performance on both static and neuromorphic datasets. In this revision, these results and the corresponding analysis are added to **Section 4.4**.
>
> **Cross-dataset transfer performance of architectures searched by DSAS:**
>
> | Original Dataset | Target Dataset | Accuracy (%) | Parameter (M) |
> | ---------------- | -------------- | ------------ | ------------- |
> | Tiny-ImageNet    | CIFAR-10       | 96.38        | 27.17         |
> | ImageNet         | CIFAR-10       | 94.02        | 15.50         |
> | CIFAR-10         | Tiny-ImageNet  | 65.41        | 38.72         |
> | CIFAR10-DVS      | CIFAR-10       | 92.97        | 13.84         |
> | CIFAR10-DVS      | DVS128-Gesture | 95.83        | 13.83         |

---

> ### Author Response · Authors · 2025-11-22
> **Response to Reviewer yxtV [Part 2/2]**
>
> > **Since the parameter spaces of ANNs and SNNs differ, it remains unclear how the TGS metric can meaningfully capture gradient alignment between teacher and student networks.**
>
> We thank your for the comment. Although ANN and SNN parameter spaces differ, TGS measures functional gradient alignment rather than exact parameter correspondence. This mechanism can be achieved by two stages. In stage 1, ANN gradients are summarized into statistics (mean and standard deviation) and used to adjust the shape of the surrogate gradient of SNN (see **Appendix D**). This can align the magnitude and distribution of gradients between teacher and student. In stage 2, TGS computes similarity using normalized Gram matrices of gradients and capturing class-wise gradient relationships. Thus, TGS effectively quantifies how similarly the ANN teacher and SNN student would update their outputs, enabling effective knowledge transfer despite heterogeneous network architectures.
>
> > **Some implementation details are missing, such as the surrogate gradient function used for training the searched SNN, and the event data representations for CIFAR10-DVS and DVS128-Gesture (e.g., whether voxel grids are used).**
>
> We are sorry for not including these details in the initial version. First, the Sigmoid function is selected as the surrogate gradient function in this study. Moreover, the event-to-frame integrating method [4] is used for preprocessing neuromorphic data representations, which is implied by the SpikingJelly [5] tool package. Specifically, the event-to-frame integrating method is a technique for representing event data that accumulates discrete events from an event camera within fixed time windows or fixed event numbers. By aggregating all positive and negative events at their corresponding pixel locations during each interval, the dense image frame is converted to be suitable for neural network processing. These details are added to **Appendix E** (Lines 594-597).
>
> > **Typo: Line 240 — "SNN Teacher" should be "SNN Student".**
>
> We sincerely apologize for this oversight and thank the reviewer for pointing it out. In this revision, we have corrected this typo in Line 245 (previous Line 240).
>
> **References**
>
> [1] Spikformer: When spiking neural network meets Transformer[C]. ICLR, 2023.
>
> [2] Auto-Spikformer: Spikformer architecture search[J]. Frontiers in Neuroscience, 2024.
>
> [3] AutoST: Training-free neural architecture search for spiking transformers[C]. ICASSP, 2024.
>
> [4] Incorporating learnable membrane time constant to enhance learning of spiking neural networks[C]. ICCV, 2021.
>
> [5] SpikingJelly: An open-source machine learning infrastructure platform for spike-based intelligence[J]. Science Advances, 2023.

---

### Author Response · Authors · 2025-11-22
**General Response**

We wholeheartedly thank all reviewers for the great effort and the meticulous review. The comments and suggestions are truly impressive and helpful to enhance our paper. In response to the comments and suggestions, we have made the following updates in this revision:

+ The experiments on the Transformer-based search space are added in **Section 4.4**, in order to verify the effectiveness of DSAS on broader search space. [yxtV] [GNxc] [8Fkb]
+ The experiments and analysis on cross-dataset generalization are added in **Section 4.4** to justify the transferability of the searched architectures. [yxtV]
+ The implementation details of the experiments are presented in **Appendix E** to improve the clarity of this study. [yxtV]
+ The typo in line 240 has been fixed. [yxtV]
+ The overall cost include pretraining ANN teacher are updated in **Section 4.4** to provide a more fair comparison. [5xcB]
+ The methodology for energy evaluation are added in **Appendix H** to enhance the clarity of the evaluation of power consumption. [5xcB]
+ The experiments about different search strategy without ANN pre-trained weights are added in **Section 4.4** and **Appendix I** to justify the effectiveness of our design on the teacher-based search. [8h8x]
+ The experiments on the random search baseline are presented in **Section 4.4** and **Appendix I** to demonstrate the effectiveness of the search strategy based on evolutionary computation. [8h8x]
+ The grammatical and expression errors have been fixed, including "DSAS" abbreviation in **Abstract**, acronyms of captions in **Section 2.1**, **2.2**, **3.2**, and **3.3**, acronyms of dataset names in line 353, and the typos in 148 and Figure 1. [8h8x]
+ The theoretical insights are added in **Appendix C** to enhance the contributions to SNN optimization. [8Fkb]
+ The clarification and modification of the expression details of the parameter K in TGS in lines 217-219, the definition of "KD", and the typos in line 187 and line 238. [8Fkb]
+ The experiments and analysis about the hyperparameter tuning method in **Section 4.4** to justify the merit of DSAS. [8Fkb]
+ The experiments and discussions on the optimization of spiking neuron type in **Section 4.4** to show the potential expansion of DSAS. [8Fkb]

Please note that the revised contents are highlighted in red in this version for the reviewers' convenience.

---

### Meta-Review · Area_Chair_fn3o · 2026-01-02

**Summary:**

This paper proposes DSAS, a training-free spiking neural architecture search method designed to distill high-performance SNN student models under low timesteps using pre-trained ANN teachers. Although the authors provided extensive revisions and additional experiments in response to reviewers, several core concerns regarding theoretical depth, methodological novelty, and practical scalability remain unresolved. The paper’s contributions are deemed insufficient for acceptance at ICLR 2026.

Summary of key reviewer concerns supporting the decision:

1. Lack of Theoretical Novelty and Depth

Reviewers noted that the proposed metrics—Multi-layer Activation Similarity (MAS) and Threshold-guided Gradient Similarity (TGS)—closely resemble existing techniques such as Representational Similarity Analysis and relational knowledge distillation, without offering clear theoretical advancement.

In particular, TGS does not adequately address the fundamental issue of surrogate gradient approximation errors in SNNs, making its claimed benefit of gradient alignment appear speculative and insufficiently justified.

2. Limited Generality and Scalability

While the authors extended experiments to Transformer-based search spaces, reviewers questioned the scalability of DSAS to large-scale models (e.g., recent foundation SNNs with tens of billions of parameters).

At such scales, neural architecture search becomes computationally prohibitive and may lack a meaningful search space for SNNs, reducing the practical relevance of the method for bridging the performance gap between large ANNs and SNNs.

3. Insufficient Experimental Rigor and Comparison

The advantages of DSAS over simpler baselines—such as random search, heuristic hyperparameter tuning, or training-free zero-cost proxies—were not convincingly demonstrated in several experimental settings.

Although cross-dataset generalization and teacher-network ablation studies were added, concerns remained about whether performance gains stemmed primarily from stronger teacher models rather than the proposed search mechanism.

4. Presentation and Clarity Issues

The initial submission contained multiple typographical errors, undefined acronyms, unclear notations, and incomplete figure captions. Although revised, these issues initially undermined the paper’s readability and precision.

Overall, despite the authors’ considerable effort in the rebuttal, the paper falls short in establishing significant theoretical innovation, general applicability, and experimental conclusiveness. Therefore, the decision is to reject the submission.

**Reviewer Concerns:**

Concerns Addressed in the Rebuttal
The authors responded constructively to several reviewer concerns by adding experiments and clarifications:

1. Extension beyond CNNs (Reviewers yxtV, GNxc)

The authors added experiments on Transformer-based search spaces (Spikformer), showing competitive results on CIFAR-10 and CIFAR-100.

2. Cross-dataset generalization (Reviewer yxtV)

New experiments evaluated transferability of searched architectures across datasets (e.g., Tiny-ImageNet to CIFAR-10), showing reasonable generalization.

3. Implementation and energy evaluation details (Reviewers yxtV, 5xcB)

Clarified surrogate gradient function (Sigmoid), event-to-frame preprocessing for neuromorphic datasets, and provided detailed energy calculation formulas in the appendix.

4. Search cost reporting (Reviewer 5xcB)

Added total design cost (including teacher pretraining) for fairer comparison with existing methods.

5. Comparison with alternative search strategies (Reviewer 8h8x)

Included experiments comparing DSAS to random search, evolutionary multi-objective search, and zero-cost proxies, demonstrating DSAS’s superiority in several settings.

6. Writing and presentation issues (Reviewers 8h8x, 8Fkb)

Corrected typos, acronym usage, figure labels, and clarified parameter definitions.

7. Teacher model selection justification (Reviewers GNxc, 8Fkb)

Provided reasoning for using PyramidNet and conducted ablations with ResNet teachers, showing DSAS works robustly across teacher architectures.

8. Neuron type selection extension (Reviewer 8Fkb)

Preliminary experiments showed DSAS can jointly search architecture and spiking neuron type (IF, LIF, PLIF, ELIF), with competitive results.

Outstanding and Unresolved Concerns
Despite the authors’ efforts, several critical issues remain:

1. Theoretical Insufficiency and Lack of Novelty (Reviewers 8Fkb, yxtV, 5xcB)

MAS is still perceived as a straightforward adaptation of relational similarity metrics to SNNs, without significant theoretical innovation.

TGS fails to address the fundamental mismatch between surrogate gradients in SNNs and true gradients in ANNs. The rebuttal did not provide theoretical or empirical analysis of how TGS mitigates surrogate gradient approximation errors.

The theoretical justifications added in Appendix C were deemed insufficient and, in some cases, mathematically questionable (e.g., claims about norm alignment under MAS).

2. Scalability to Large/Foundation Models (Reviewer 8Fkb)

The authors acknowledged the challenge but did not convincingly argue for the applicability of NAS in large-scale ANN-to-SNN distillation (e.g., for models like SpikingBrain with 76B parameters). The method’s relevance in the regime where distillation matters most remains in doubt.

3. Core Motivation and Necessity of NAS in ANN-to-SNN Distillation (Reviewer 8Fkb)

A key philosophical concern persists: whether architectural search meaningfully addresses the fundamental bottlenecks of SNNs (spike dynamics, surrogate gradient errors), which are algorithmic and structural rather than purely architectural.

4. Experimental Rigor and Fairness (Reviewers GNxc, 5xcB)

Doubts remain about whether performance gains stem from the search mechanism or from stronger teacher models. While teacher ablation studies were added, a clear isolation of the NAS contribution is still lacking.

Comparison to heuristic hyperparameter tuning and simpler baselines still does not conclusively demonstrate that DSAS offers a substantial advantage beyond incremental improvement.

5. Clarity on Neuron-Type Search Results (Reviewer 8Fkb)

Although the authors showed that neuron-type selection can be integrated, they did not provide concrete examples of which neuron types were selected under different conditions or why certain mixtures performed better, leaving the mechanism somewhat opaque.

Overall Assessment
The authors have made commendable efforts to address practical and empirical concerns (e.g., new datasets, architectures, implementation details). However, the core issues regarding theoretical contribution, methodological necessity, and scalability significance remain unresolved.

Therefore, based on the remaining fundamental weaknesses in theoretical grounding and methodological justification, the paper is recommended for rejection.

**Reviewer Scores:**

Reviewer yxtV
Original: 6 (marginally above threshold)
Addressed: Transformer experiments, cross-dataset transfer, implementation details.
Unresolved: Weak theoretical grounding for TGS, limited novelty of MAS.
Estimated change: 6 (unchanged)
Reason: Empirical improvements acknowledged, but theoretical contribution remains questionable.

Reviewer 5xcB
Original: 4 (marginally below threshold)
Addressed: Total cost reporting, energy evaluation methodology.
Unresolved: MAS similarity to prior work, incremental contribution.
Estimated change: 4 (unchanged)
Reason: Methodological novelty still insufficient.

Reviewer GNxc
Original: 4 (marginally below threshold)
Addressed: Transformer experiments, teacher-network ablations.
Unresolved: Whether gains stem from teacher strength, scalability concerns.
Estimated change: 4 (unchanged)
Reason: Necessity and generalizability not convincingly demonstrated.

Reviewer 8h8x
Original: 4 (marginally below threshold)
Addressed: Random search comparison, writing corrections.
Unresolved: Temporal dynamics ignored in MAS, gradient alignment issues in TGS.
Estimated change: 4 (unchanged)
Reason: Core algorithmic limitations remain unaddressed.

Reviewer 8Fkb
Original: 4 (marginally below threshold)
Addressed: Transformer experiments, neuron-type search feasibility.
Unresolved: Weak theoretical justification, poor scalability to large models, questionable necessity of NAS in distillation.
Estimated change: 4 (unchanged)
Reason: Foundational concerns not alleviated.

Summary
Post-rebuttal, most reviewers would likely retain the scores. This supports the rejection decision—the paper still falls short of ICLR high standards in theoretical novelty, methodological necessity, and long-term impact.

---

### Decision · Program_Chairs · 2026-01-26

Reject